# Functional diversity among cardiolipin binding sites on the mitochondrial ADP/ATP carrier

Nanami Senoo [ID] [1,2], Dinesh K Chinthapalli[3,4], Matthew G Baile[1], Vinaya K Golla [ID] [5,6], Bodhisattwa Saha[3], Abraham O Oluwole [ID] [3,4], Oluwaseun B Ogunbona [ID] [1], James A Saba[1], Teona Munteanu[1], Yllka Valdez[1], Kevin Whited[1], Macie S Sheridan[1,2], Dror Chorev [ID] [3], Nathan N Alder[5], Eric R May[5], Carol V Robinson[3,4] & Steven M Claypool [ID] [1,2,7 ✉]

## Abstract

**Lipid-protein interactions play a multitude of essential roles in membrane homeostasis. Mitochondrial membranes have a unique lipid-protein environment that ensures bioenergetic efficiency. Cardiolipin (CL), the signature mitochondrial lipid, plays multiple roles in promoting oxidative phosphorylation (OXPHOS). In the inner mitochondrial membrane, the ADP/ATP carrier (AAC in yeast; adenine nucleotide translocator, ANT in mammals) exchanges ADP and ATP, enabling OXPHOS. AAC/ANT contains three tightly bound CLs, and these interactions are evolutionarily conserved. Here, we investigated the role of these buried CLs in AAC/ANT using a combination of biochemical approaches, native mass spectrometry, and molecular dynamics simulations. We introduced negatively charged mutations into each CL-binding site of yeast Aac2 and established experimentally that the mutations disrupted the CL interactions. While all mutations destabilized Aac2 tertiary structure, transport activity was impaired in a binding site-specific manner. Additionally, we determined that a disease-associated missense mutation in one CL-binding site in human ANT1 compromised its structure and transport activity, resulting in OXPHOS defects. Our findings highlight the conserved significance of CL in AAC/ANT structure and function, directly tied to specific lipid-protein interactions.**

**Keywords** Lipid–Protein Interaction; Mitochondria; Membrane Transport; Oxidative Phosphorylation; Cardiolipin
**Subject Categories** Metabolism; Organelles

## Introduction

Biological membranes consist heterogeneously of proteins and lipids. Lipids can serve as solvents, substrates, and regulatory cofactors for membrane proteins, which in turn modulate the structural and functional properties of membrane proteins (Leventol and Lyman, 2023; Wu et al, 2016). The biological significance of membrane lipid–protein interactions has been reported in several studies, such as for rhodopsin, G protein-coupled receptors, and ion channels (Leventol and Lyman, 2023). Yet the mechanisms by which lipid–protein interactions influence membrane protein structure and function remains at an early stage in our understanding, mainly due to limited methodologies available to detect specific lipid–protein interactions and define the functionality of the interactions (Leventol and Lyman, 2023).

Mitochondria contain two membranes, the outer mitochondrial membrane (OMM) and the inner mitochondrial membrane (IMM). While the OMM compartmentalizes the organelle by forming its outer edge, the IMM forms infoldings called cristae which concentrate the bioenergetic components required for oxidative phosphorylation (OXPHOS) (Palade, 1953; Vogel et al, 2006; Wilkens et al, 2013; Iovine et al, 2021). Cardiolipin (CL) is a phospholipid enriched in mitochondria, especially in the IMM (Horvath and Daum, 2013). Consisting of two phosphate head-groups and four acyl chains, the unique structure of CL supports and optimizes many biological events occurring in mitochondria (Acoba et al, 2020). Focused on the OXPHOS machinery, the main documented roles of CL include physically associating with individual components (Ruprecht et al, 2014; Nury et al, 2005; Pebay-Peyroula et al, 2003; Beyer and Klingenberg, 1985; Coleman et al, 1990; Fiermonte et al, 1998; Gomez and Robinson, 1999b, 1999a; Kadenbach et al, 1982; Lange et al, 2001; Sedlák and Robinson, 1999; Shinzawa-Itoh et al, 2007; Rathore et al, 2019; Hartley et al, 2019) and stabilizing their higher order assembly into supercomplexes (Pfeiffer et al, 2003; Zhang et al, 2011, 2002). Loss of CL diminishes energetic capacity in multiple organisms (Claypool et al, 2008; Baile et al, 2014; Jiang et al, 2000; Sustarsic et al,

[1]Department of Physiology, Johns Hopkins University School of Medicine, Baltimore, MD 21205, USA. [2]Mitochondrial Phospholipid Research Center, Johns Hopkins University School of Medicine, Baltimore, MD 21205, USA. [3]Physical and Theoretical Chemistry Laboratory, University of Oxford, Oxford OX1 3QU, UK. [4]Kavli Institute for Nanoscience Discovery, Oxford OX1 3QU, UK. [5]Department of Molecular and Cell Biology, University of Connecticut, Storrs, CT 06269, USA. [6]Department of Cell Biology, University of Virginia School of Medicine, Charlottesville, VA 22903, USA. [7]Department of Genetic Medicine, Johns Hopkins University School of Medicine, Baltimore, MD 21205, USA. ✉E-mail: sclaypo1@jhmi.edu

2018; Lee et al, 2022). This implies that conserved CL–protein interactions serve critical and functionally important roles for the OXPHOS machinery. Although the significance of CL in mitochondrial biology has been increasingly accepted, mechanistic details of how this lipid supports membrane protein structure and function are still lacking.

The ADP/ATP carrier (AAC in yeast; adenine nucleotide translocator, ANT in humans) belongs to the largest solute carrier family, SLC25, which includes 53 members in humans. SLC25 family members are mostly located in the IMM where they collectively mediate the flux of a wide variety of solutes, such as ions, cofactors, and amino acids, across the otherwise impermeable IMM (Ogunbona and Claypool, 2019). AAC is embedded in the IMM and transports ATP out of and ADP into the mitochondrial matrix in a 1:1 exchange. This function is required for OXPHOS and needed to make ATP synthesized in the matrix bioavailable to the rest of the cell. Of the three isoforms in yeast, Aac2 is the most abundant and only isoform required for OXPHOS (Lawson et al, 1990). AAC alters its conformation during the transport cycle. Inhibitors are available to trap the carrier into two extreme conformational states: carboxyatractyloside (CATR) locks the carrier in the cytosolic open state (c-state) and bongkrekic acid (BKA) locks it in the matrix open state (m-state) (Ruprecht et al, 2014, 2019).

First demonstrated via $^{31}$-P NMR (Beyer and Klingenberg, 1985) and subsequently confirmed in all of the solved crystal structures of AAC locked in the c-state with CATR, including yeast Aac2 (Ruprecht et al, 2014) and bovine ANT1 (Nury et al, 2005; Pebay-Peyroula et al, 2003), AAC contains three tightly bound CL molecules. In the m-state structure of AAC with BKA, one of the three CLs remains bound to AAC whereas the second and third are lost (Ruprecht et al, 2019), suggesting that the affinities of the CL-binding sites are conformation-sensitive. Protein thermostability assays have shown that CL enhances the stability of yeast Aac2 (Crichton et al, 2015). Recent molecular dynamics (MD) simulation studies indicate that CL affects the structure of AAC/ANT (Montalvo-Acosta et al, 2021; Yi et al, 2022). Our previous study further emphasized the diverse structural roles provided by CL: CL supports Aac2 tertiary structure and its transport-related conformation (Senoo et al, 2020). In addition, through a distinct mechanism, CL regulates the association of Aac2 with other OXPHOS components (Claypool et al, 2008; Senoo et al, 2020).

The fact that the three CLs are bound to AAC in an evolutionarily conserved manner underscores the functional significance of these lipid–protein interactions; however, molecular-level insight into if and how these CL molecules contribute to AAC structure and function has not been experimentally interrogated. In this study, we engineered a series of mutations to disrupt specific CL interactions in yeast Aac2 and experimentally verified that the lipid–protein interactions are structurally important. Interestingly, while each bound CL promotes Aac2 transport, one lipid–protein interaction is functionally essential. Finally, we provide evidence that a pathologic mutation in *ANT1* tied to human disease is structurally and functionally compromised due to the disruption of one of the three conserved tightly bound CLs. To our knowledge, this is the first documented disease-associated missense mutation whose defect derives from a disturbed interaction between a mutant protein and a specific lipid.

# Results

## Engineered CL-binding Aac2 mutants

The solved crystal structures of AAC in the c-state all include three tightly bound CL molecules (Ruprecht et al, 2014; Nury et al, 2005; Pebay-Peyroula et al, 2003). In each CL-binding pocket, the phosphate headgroup of CL engages the backbone peptide. Aiming to disrupt the lipid–protein interaction via electrostatic repulsion, we introduced negatively charged amino acids in the immediate vicinity of each CL-binding pocket of yeast Aac2 (Fig. 1A). The mutations were designed at symmetrically related locations per pocket. One type of mutation is on even-numbered transmembrane helices (I51 for pocket 1, L155 for pocket 2, M255 for pocket 3) and another is on the matrix loops immediately preceding matrix helices (G69 for pocket 1, G172 for pocket 2, G267 for pocket 3) (Fig. 1B). Additional mutations were designed for pocket 2 (N90) and pocket 3 (R191, L194) in a contiguous region of the matrix helices and even-numbered transmembrane helices (Fig. 1B). Predicted to prevent the engagement of the associated CL phosphates like the other mutants, it is of note that N90 and R191 are close to several residues involved in substrate binding related to the AAC transport cycle as recently shown (Mavridou et al, 2022). Every mutant was expressed at an equal level to WT Aac2 as re-introduced into the *aac2Δ* parental cells (Fig. 1C). While every mutant was able to grow on respiratory media (YPEG) at 30 °C, in which cellular energy production relies on OXPHOS, the respiratory growth capacity varied (Fig. 1D). The respiratory growth defects of *crd1Δ* and some of the Aac2 mutants was exacerbated at elevated temperature (37 °C) (Fig. 1D), suggesting the involvement of CL and potentially CL-Aac2 in yeast adaptation to stress.

Aac2 contains six membrane-passing domains with the N- and C- termini exposed to the intermembrane space (Pebay-Peyroula et al, 2003). When the outer membrane was removed by swelling, proteinase K treatment erased the IMS-exposed N-terminus of Aac2 which was detected by a monoclonal antibody recognizing this peptide (Panneels et al, 2003). With the possible exception of G172E, each Aac2 mutant presented the same pattern as WT Aac2 (Fig. 1E). This data demonstrates that the inserted negatively charged amino acids do not disrupt the topology of the mutant Aac2 proteins in the IMM.

## Specific lipid–protein interactions are disrupted in CL-binding Aac2 mutants

To identify the lipid–protein interactions of Aac2, we established a native mass spectrometry (MS) approach as our prior study successfully detected the interactions between bovine ANT1 and CLs (Chorev et al, 2018). We introduced a Flag-tag onto the N-terminus of WT and mutant Aac2 and purified FlagAac2 from mitochondria using the detergent undecylmaltoside (UDM). Before MS analysis, the purified protein was rapidly buffer-exchanged into lauryldimethylamine oxide (LDAO). The growth phenotype of the Flag-tagged mutants was the same as untagged mutants (Appendix Fig. S1A; Fig. 1D), indicating the Flag-tag is functionally silent. We noted that the use of UDM during purification was structurally destabilizing to WT Aac2; on blue native-PAGE, a smear above ~200 kDa was seen (Appendix Fig. S1B) instead of stabilized Aac2

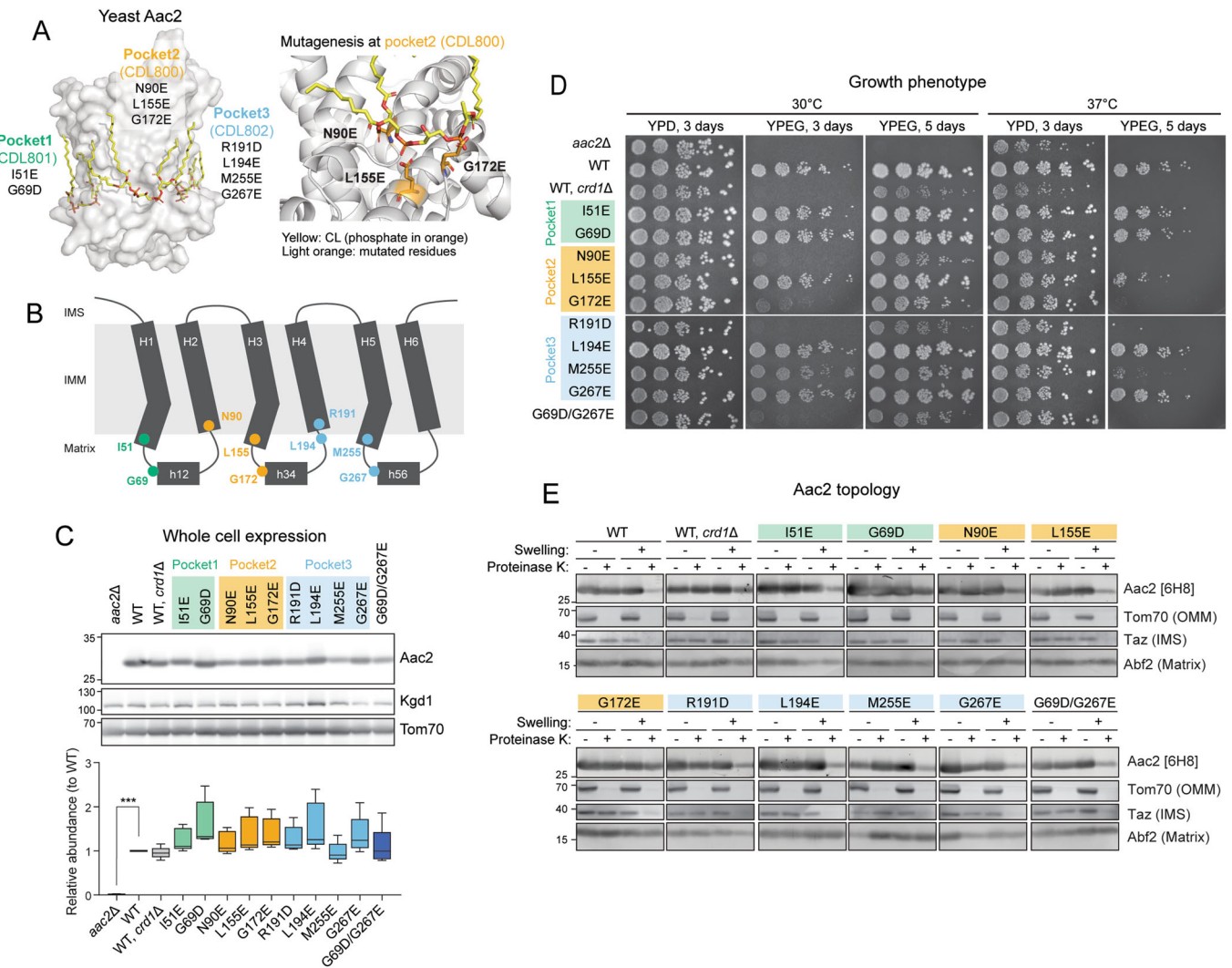

**Figure 1. Generation and characterization of yeast Aac2 CL-binding mutants.**

(**A**) Strategy to disrupt CL-binding sites in Aac2. Aac2 was modeled onto bovine ANT1 (PDB ID: 2C3E) using SWISS-MODEL. Negatively charged amino acids introduced (light orange) into Aac2 CL-binding motifs face toward the CL phosphate headgroup (orange). (**B**) Schematic representation showing the positions of designed mutations in Aac2. (**C**) Expression of WT and mutant Aac2 was detected in whole cell extracts by immunoblot; Kgd1 and Tom70 served as loading controls ($n = 5$, biological replicates). Data were shown as box-whisker plots with the box extended from 25th to 75th percentiles and the whiskers indicating the min to max range. Significant difference was obtained by one-way ANOVA with Dunnett's multiple comparisons test (vs. WT) ***$p < 0.001$. (**D**) Growth phenotype of Aac2 CL-binding mutants. Serial dilutions of indicated cells were spotted onto fermentable (YPD) and respiratory (YPEG) media and incubated at 30 or 37 °C for 3–5 days ($n = 3$, biological replicates). (**E**) Membrane topology of WT and mutant Aac2. Isolated mitochondria were osmotically ruptured and treated with or without proteinase K as indicated. Aac2 N-terminus was detected by a monoclonal antibody 6H8 recognizing the first 13 amino acids MSSNAQVKTPLPP ($n = 6$, biological replicates). IMS intermembrane space, IMM inner mitochondrial membrane, OMM outer mitochondrial membrane. In (**C–E**), representative images from the indicated replicates are shown. Source data are available online for this figure.

monomer at ~140 kDa (Senoo et al, 2020). Given this, we treated mitochondria with CATR before UDM-solubilization which indeed preserved the major ~140 kDa Aac2 monomer, similar to what is observed following digitonin extraction (Appendix Fig. S1B). Consistently, CATR pretreatment improved the purification efficiency of FlagAac2.

For WT Aac2 isolated from CL-containing mitochondria, native MS displayed charge state series that correspond in mass to Aac2 in apo and in multiple ligand-bound forms (Fig. 2A). A typical spectrum displayed a series of charge states in the mass range $m/z$ 3500–6000, with 8+ being the most abundant. Well-resolved

protein–ligand complexes can be assigned to Aac2+CATR, Aac2 + 1CL, Aac2 + 2CL, Aac2+CATR + 1CL, Aac2+CATR + 2 CL, and Aac2+CATR + 3CL (Fig. 2A). Since the binding of CATR locks Aac2 in the c-state, the native MS dataset indicated that up to 3 CLs molecules are bound to Aac2 when stabilized in the c-state, consistent with the reported crystal structures of Aac2 with CATR (Ruprecht et al, 2014; Nury et al, 2005; Pebay-Peyroula et al, 2003). To understand the relative affinity of each CL molecule bound to Aac2, we selected the 8+ charge state of the Aac2+CATR + 3CL complex for tandem MS. We were able to dissociate the first and second CL molecule from this complex upon activation with

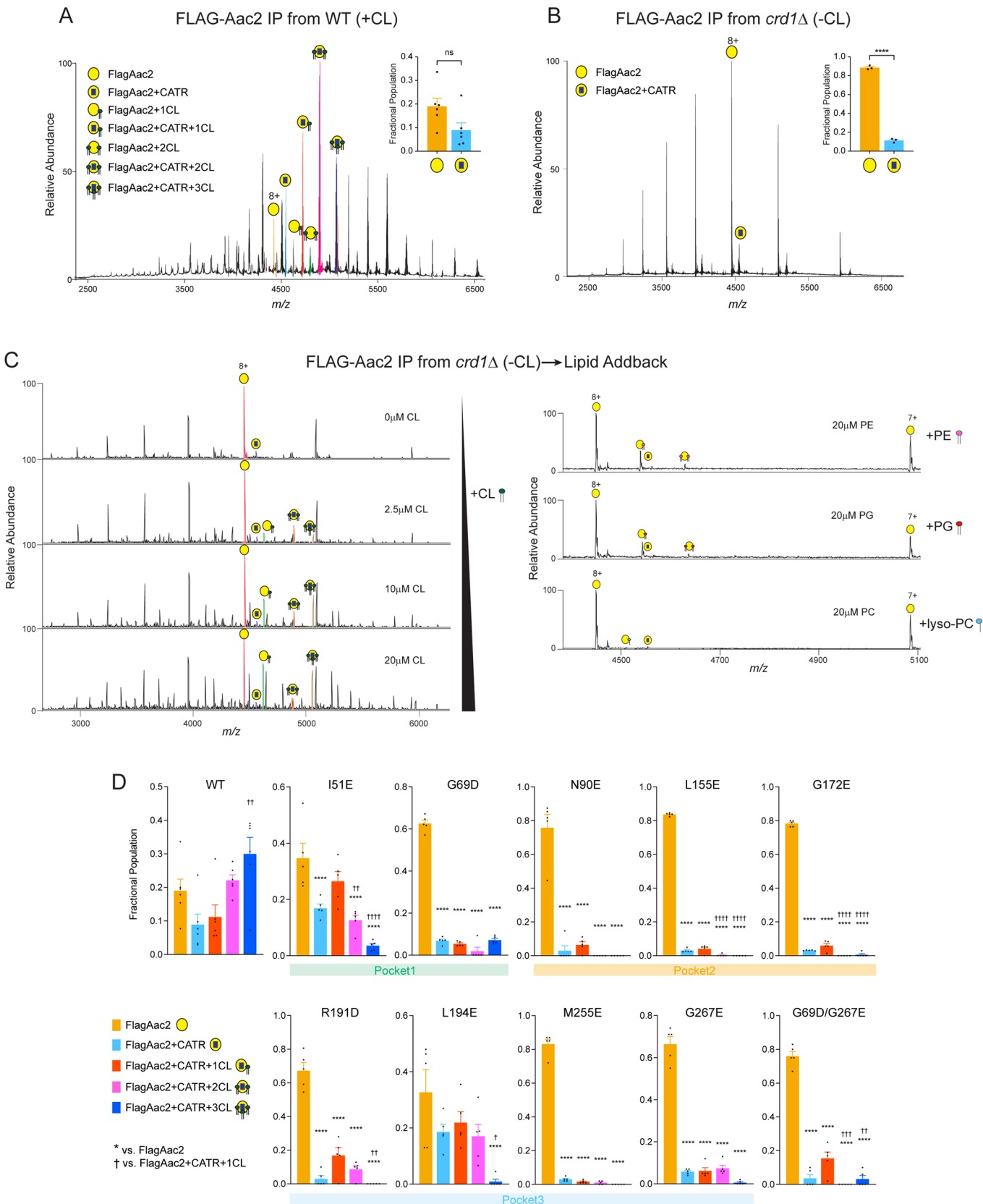

**Figure 2. Native mass spectrometry analysis to detect lipid–protein interaction of Aac2.**

(A) FlagAac2 affinity purified from WT mitochondria was associated with CATR and up to three CL molecules. (B) FlagAac2 affinity purified from *crd1Δ* mitochondria which lack CL did not co-purify other phospholipids. Insets in (A) and (B) show the fractional population of FlagAac2 relative to FlagAac2+CATR when immunoprecipitated from WT and *crd1Δ* mitochondria, respectively. Mean with SEM ($n = 3$–6: 3 biological replicates with 1–2 technical replicates). Significant differences as determined by Student's *t*-test indicated (****$p < 0.0001$). (C) CL, PE, PG, or lyso-PC was added to FlagAac2 purified from *crd1Δ* mitochondria at the indicated concentrations. (D) Fractional population of FlagAac2, FlagAac2+CATR alone or associated with one to three CL molecules was determined for WT and Aac2 CL-binding mutants. Mean with SEM ($n = 5$–6: three biological replicates with 1–2 technical replicates). Significant differences were obtained by one-way ANOVA with Tukey's multiple comparisons test (* vs. FlagAac2; † vs. FlagAac2+CATR + 1CL); */† $p < 0.05$, **/†† $p < 0.01$, ***/††† $p < 0.001$, ****/†††† $p < 0.0001$. Source data are available online for this figure.

150 and 200 V in the high-energy collisional dissociation (HCD) cell (Appendix Fig. S2). The sequential loss of ~1400 Da corresponds to the average mass of CL (Appendix Fig. S2) thus confirming our mass spectral assignments. Upon further increase of the HCD energy to 270 V, all three CLs bound to the Aac2+CATR + 3CL complex dissociated, yielding peaks assigned to apo Aac2 and Aac2+CATR (Appendix Fig. S2). The relatively high collisional voltages required to dissociate CLs suggest high affinity binding interactions with Aac2 (Appendix Fig. S2). The peaks assigned to apo- and CATR-bound Aac2 consistently exhibit a more homogeneous mass distribution than the CL-bound complexes, an indication of considerable variation in the acyl chains of bound CL molecules. Accordingly, the acyl chains of CL co-purified with Aac2 were heterogeneous (CL64:4, CL 66:4, CL68:4) (Appendix Fig. S3) and mirrored the natural CL profile of yeast (Baile et al, 2014; Poveda-Huertes et al, 2021). Due to this heterogeneity, we could only predict that CL68:4 contained 16:1 and 18:1 (estimated as 16:1-16:1-18:1-18:1) as the chain lengths matched with those obtained via MSMS spectra (Appendix Fig. S3).

For the WT Aac2 purified from *crd1Δ* mitochondria which lack CL, the native MS yielded ions that correspond in mass to apo and CATR-bound Aac2 only; neither CL nor any other phospholipid including the CL precursor, phosphatidylglycerol (PG) which accumulates in the *crd1Δ* strain (Baile et al, 2014), were co-purified with Aac2 (Fig. 2B). We also examined the capacity of Aac2 in *crd1Δ* to recoup CL binding by adding CL exogenously. The interactions of added CL to Aac2 occurred in a titrated manner: minimal CL binding was achieved when incubated with 1 μM CL, and 2–3 CLs were bound at 2.5 μM (Fig. 2C). By contrast, exogenously added PG, lyso-phosphatidylcholine (lyso-PC), and phosphatidylethanolamine (PE) only bound to Aac2 at relatively higher concentrations (Fig. 2C). Note that an increase in number of charge state distributions in *crd1Δ* mitochondria (Fig. 2B) indicated that Aac2 is denatured or partially unfolded due to the absence of CL. Given this, the binding of PG, lyso-PC, and PE phospholipids at high concentrations are likely nonspecific. These results demonstrate that Aac2 lipid binding is highly specific to CL and that other lipid classes are unable to structurally replace CL as contained in the IMM or added exogenously.

To account for the relative strengths/extent of disruption of CL-Aac2 interactions with our mutational strategy, the mass spectra were deconvoluted to obtain the fractional populations of each protein species (Fig. 2D; see methods). For WT Aac2, the most abundant species was Aac2+CATR + 3CL (Fig. 2D), as expected for the CATR-bound form (Ruprecht et al, 2014; Nury et al, 2005; Pebay-Peyroula et al, 2003). For all mutants, the populations of apo Aac2 were the highest with a dramatic reduction in the populations of Aac2+CATR + 3CL (Fig. 2D), demonstrating the feasibility of our mutational strategy. Though this is evident in all the mutants, the extent of disruption of Aac2+CATR + 3CL was less in the mutants generated in pocket 1 (I51E and G69D) and double mutant (G69D/G267E). Prominent destabilization of CL-protein interactions was bestowed by the mutants of pocket 2 wherein fractional populations of Aac2 + CATR + 2CL were also significantly reduced. This disruption of Aac2 + CATR + 2CL was also seen in the double mutant (G69D/G267E) (Fig. 2D). Interactions of Aac2 + CATR + 1CL were retained in all the mutants and with two exceptions (G69D and G69D/G267E) trended to be higher than the three CLs bound form. Although the relative abundances of the CATR-bound Aac2 with CLs species were significantly decreased in all the mutants except I51E and L194E, none of the mutants disrupted all three CATR-stabilized Aac2–CL interactions completely. Naively, we predicted that the population of CATR-bound Aac2 + 3 CLs would be the most reduced for each mutant as the inserted mutation targeted only one CL-binding pocket; in principle, the other two pockets should be intact and CL-binding competent. Although this pattern was obtained for I51E and L194E, the majority of mutants displayed more broadly disrupted CL-binding. This could indicate that ablation of any single tightly bound CL weakens the tertiary stability of mutant Aac2 and that the application of additional structurally perturbing agents, such as the harsh detergents used for native MS sample preparation, causes mutants to denature and thus release most of the bound CLs.

Since the charge state series of a protein in a mass spectrum is related to the exposed surface area of the protein, it is an exquisitely sensitive proxy for the overall fold of the protein (Hall and Robinson, 2012). Evidence for this relationship comes from ion mobility experiments which show that in the gas phase, higher charge states correspond to more unfolded proteins while lower charge states represent more compact structures (Hanauer et al, 2007; Clemmer and Jarrold, 1997). It is also evident when there is a particularly large shift in charge state when a protein is unfolded. A minor perturbation in folded structure may result in a slight shift in charge state distribution of one protonation site (Hanauer et al, 2007; Clemmer and Jarrold, 1997). With this in mind, we compared the most abundant charge state 8+ in each case (representative spectra are shown in Appendix Fig. S4). In comparison to the WT charge state distribution, there was no large shift in the charge state for all of the mutants investigated, indicating that all the mutants are folded. Several mutants, including L155E, N90E, G172E, and R191D, had slight perturbation of their charge states, implying small changes to their surface area may have occurred; however, these changes are not sufficient for the protein to be considered unfolded.

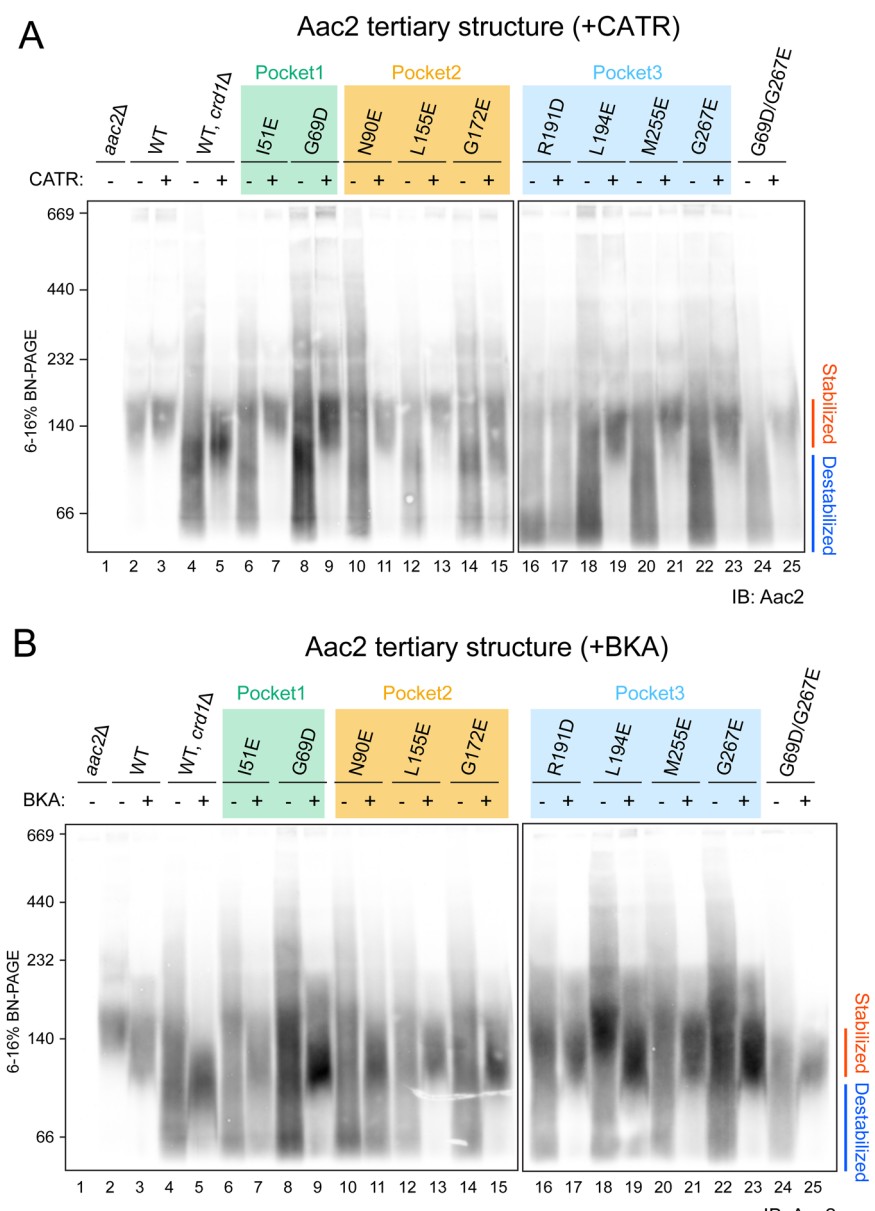

**Figure 3. CLs associated with Aac2 stabilize its tertiary structure.**

Mitochondria (100 μg) from WT and Aac2 CL-binding mutants were mock-treated or instead incubated with either 40 μM CATR (**A**) or 10 μM BKA (**B**) and then solubilized with 1.5% (w/v) digitonin. The extracts were resolved by 6 to 16% blue native-PAGE and immunoblotted for Aac2 ($n = 6$, biological replicates). Representative images from the indicated replicates are shown. Source data are available online for this figure.

## CLs directly bound to Aac2 stabilize its tertiary structure

We previously demonstrated that CL stabilizes the Aac2 tertiary structure (Senoo et al, 2020) but whether this property derived from one or more of the co-crystallized CL molecules was unresolved. To ask whether the CL molecules directly interacting with Aac2 affect its tertiary structure, we solubilized WT and mutant mitochondria with the mild detergent digitonin and resolved them on blue native-PAGE. Consistent with our previous results (Senoo et al, 2020), the majority of WT Aac2 formed a stable tertiary structure (locked at ~140 kDa) in the presence of CL (lane 2 in Fig. 3A,B). In the absence of CL (*crd1Δ*), the

Aac2 tertiary structure was destabilized and instead detected as a smear from ~230 to <67 kDa (lane 4 in Fig. 3A,B). In previous studies, we have established that the detection of this smear in CL-lacking mitochondria does not likely reflect aberrant protein interactions (Claypool et al, 2008; Senoo et al, 2020). Importantly, all Aac2 mutants were destabilized even in the presence of CL (even-numbered lanes starting from six in Fig. 3A,B); this strongly indicates that, as we hypothesized, all three bound CLs are crucial in stabilizing the Aac2 tertiary structure.

To rule out the possibility of aberrant interactions contributing to the smear seen in the CL-binding mutants, we performed FLAG

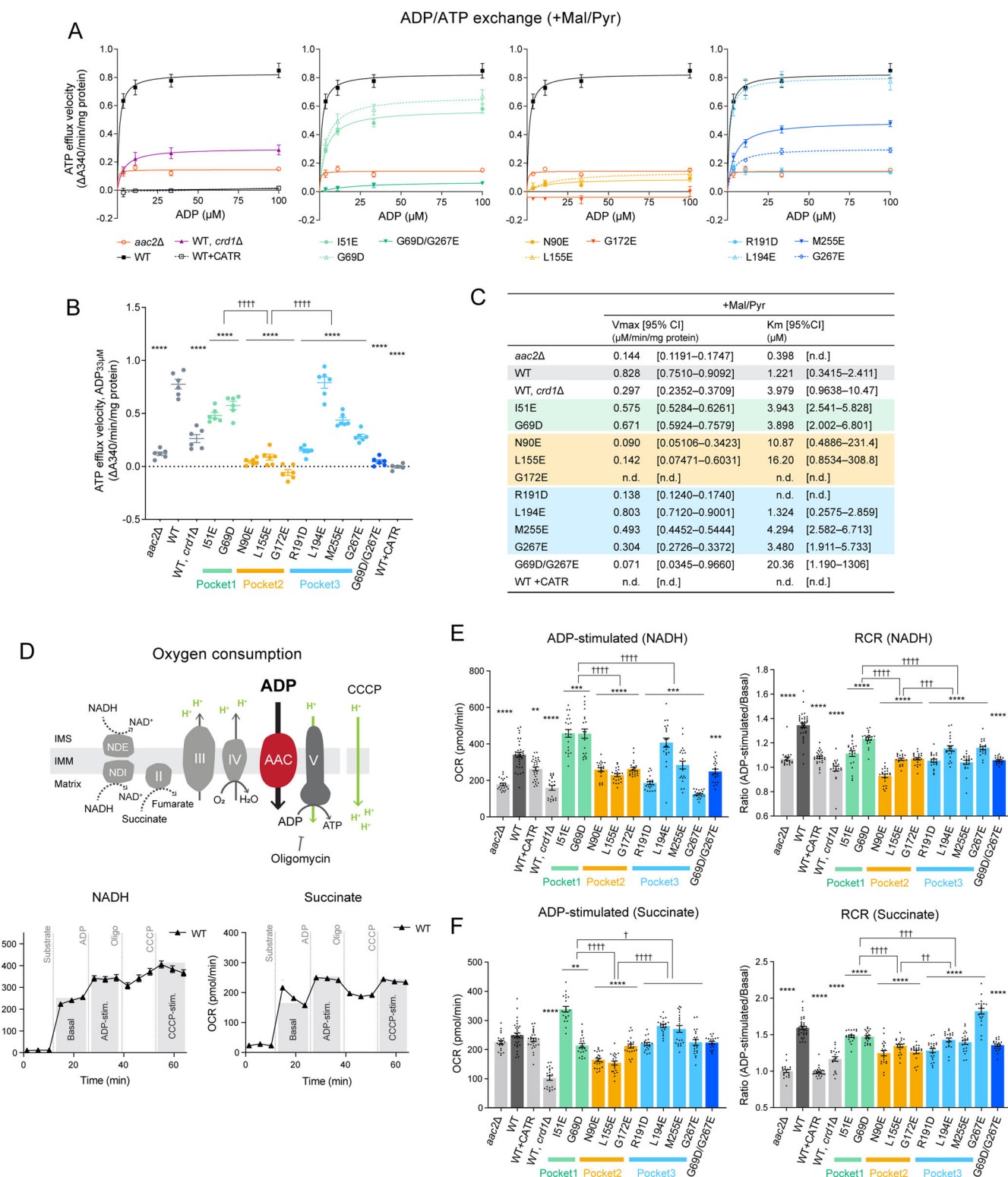

immunoprecipitation and analyzed the pattern of protein interactions by SDS-PAGE (Fig. EV1). The results showed that WT Aac2 was co-detected with multiple bands, with those ranging from 10 to 40 kDa estimated to be the interaction with respiratory complex subunits, as per our previous study (Claypool et al, 2008). The

mutants retained identical protein interaction patterns to WT, although N90E displayed relatively weaker interactions, particularly evident in the 20–30 kDa range. These results strongly suggest that mutants do not engage in aberrant protein interactions. This further underscores our interpretation that the low molecular

**Figure 4.   Distinct functional roles of the three tightly bound CLs.**

(A–C) ADP/ATP exchange: The efflux of matrix ATP was detected with isolated mitochondria as NADPH formation (A340; absorbance at 340 nm) occurring coupled with an in vitro glycolysis reaction which contained glucose, hexokinase, and glucose-6-phosphate dehydrogenase. The reaction was initiated by adding ADP. The measurement was performed in the presence of 5 mM malate and 5 mM pyruvate (+Mal/Pyr). Where indicated, WT mitochondria were treated with 5 µM CATR prior to the efflux reaction. (A) The linear part of the initial velocity for the ATP efflux was plotted and curve fitting performed by nonlinear regression (mean with SEM, $n = 6$, biological replicates). Plots of $aac2\Delta$ and WT are repeated in all panels. (B) The initial velocity following the addition of 33 µM ADP was presented as scatter plots (mean with SEM, $n = 6$, biological replicates). (C) Fitted Km and Vmax values were obtained using the Michaelis–Menten equation (mean). The range of determined values is shown in brackets (n.d., not detected). (D) Oxygen consumption rate (OCR) in isolated mitochondria with sequential injections of the respiratory substrate (NADH or succinate), ADP, oligomycin, and CCCP was measured using Seahorse XF96e FluxAnalyzer. (E, F) ADP-stimulated respiration and respiratory control ratio (RCR) of WT and Aac2 mutant mitochondria under NADH (E) and succinate (F) were plotted. RCR was obtained by dividing OCR of ADP-stimulated respiration by that of basal respiration (see also Fig. S5). WT mitochondria was treated with 50 µM CATR before the measurement. Mean with SEM, $n = 21$–35, 3–5 biological replicates with 5–7 technical replicates. Significant differences obtained by two-way ANOVA followed by Tukey's multiple comparisons test are shown as * for comparison with WT and † for comparison between pocket mutants; *$p < 0.05$, **$p < 0.01$, ***$p < 0.001$, ****$p < 0.0001$. Source data are available online for this figure.

weight smear detected following blue native-PAGE reflects the compromised tertiary stability of each CL-binding mutant.

We also tested the effects of the inhibitors CATR and BKA, which tightly lock the carrier in distinct c-state and m-state conformations, respectively (Ruprecht et al, 2014, 2019). As seen previously (Senoo et al, 2020), pretreatment of $crd1\Delta$ mitochondria with either CATR or BKA fixed the Aac2 tertiary structure (lane 5 in Fig. 3A,B), demonstrating that Aac2 is properly folded and inserted into the IMM even in the absence of CL because CATR and BKA do not bind unfolded AAC (Crichton et al, 2015). The addition of CATR pre-digitonin solubilization rescued the predominant ~140 kDa Aac2 c-state conformer of each CL-binding mutant (odd-numbered lanes starting from 7 in Fig. 3A) save one, R191D (lane 17 in Fig. 3A). Intriguingly, the addition of BKA predigitonin solubilization did rescue the predominant ~120 kDa Aac2 m-state conformer of the mutants including R191D (odd-numbered lanes starting from 7 in Fig. 3B), similar to WT Aac2 in mitochondria lacking CL (lane 5 in Fig. 3B). Taken together with the membrane topology data (Fig. 1E) and the charge states observed with MS (Appendix Fig. S4), these results demonstrate that every CL-binding Aac2 mutant is properly folded in the IMM. We conclude that interfering with the interaction of only one of the three buried CL molecules is sufficient to significantly destabilize the tertiary fold of Aac2 under blue native-PAGE conditions.

## The three tightly bound CLs are functionally distinct

To clarify if the bound CL molecules are important for Aac2 transport activity, we measured ADP/ATP exchange in isolated mitochondria from CL-binding Aac2 mutants. ADP/ATP exchange was initiated by the addition of ADP and the rate of ATP efflux was measured (De Marcos Lousa et al, 2002; Passarella et al, 1988; Hamazaki et al, 2011). The respiratory substrates malate and pyruvate were included in the reaction to establish a robust proton motive force across the IMM which is known to stimulate AAC-mediated transport (Klingenberg, 2008). The initial velocities over a range of ADP concentrations demonstrated that, as expected, WT Aac2 had the highest capacity for ATP efflux, i.e., Aac2 transport activity (Fig. 4A,B). The mutants, except L194E, had reduced Aac2 transport activity (Fig. 4A,B). The kinetics of ATP efflux was further evaluated by the Michaelis–Menten equation (Fig. 4C). Compared to WT Aac2, the mutants had higher $K_m$ values for ADP, which represents the substrate concentration at which half of the maximum reaction velocity ($V_{max}$) is reached; this suggests that

the mutants are less prone to mediate ADP/ATP exchange. There was a difference in the Aac2 transport activity across the three CL-binding pockets: the most severe defects were observed in pocket 2 mutants, followed by pocket 3 mutants with the exception of L194E. Although pocket 1 mutants retained near WT transport activity, the double pocket 1/3 mutant (G69D/G267E) showed a stronger defect in Aac2 transport activity than either corresponding single mutant. In the absence of respiratory substrates, mutant transport was much slower than when the mitochondria were energized, as expected, and the values between replicates were more variable. Despite this variability, the same trend as for energized mitochondria was still observed (Appendix Fig. S5).

Aac2-associated OXPHOS capacity was assessed by measuring oxygen consumption with isolated mitochondria. We sequentially injected a respiratory substrate (NADH or succinate) and ADP using a Seahorse FluxAnalyzer (Figs. 4D–F and EV2). ADP-stimulated oxygen consumption is dependent on ADP import across the IMM through AAC. In the presence of NADH, pocket 2 and pocket 3 mutants showed reduced ADP-stimulated respiration compared to WT Aac2 (Fig. 4E). Energized with succinate, pocket 2 mutants had reduced ADP-stimulated respiration (Fig. 4F). Similar trends were observed for each respiratory substrate following the addition of the uncoupler, CCCP (Fig. EV2). In addition, we calculated the respiratory control ratio (RCR), defined by the ratio of oxygen consumption while ADP is phosphorylated (ADP-stimulated) to that in the absence of ADP (basal), which provides insight into OXPHOS coupling and the possibility of proton leak. As expected, in the absence of ADP/ATP exchange ($aac2\Delta$ and WT + CATR), RCR was ~1 under both NADH and succinate conditions (Fig. 4E,F). With both substrates, the RCR of all pocket mutants was lower than WT; however, pocket-specific differences were detected, with RCR values decreasing for pockets 1, 3, and 2, in that order, relative to WT (Fig. 4E,F). Taken together, our data indicate that all three CL molecules bound to Aac2 regulate its transport activity but that the three buried CLs are functionally distinct: Pocket 2 has the preeminent role in supporting AAC transport activity, at least in the exchange modes tested.

## Disturbed CL-Aac2 interaction attenuates respiratory complex expression, activity, and assembly

Yeast complex IV includes three subunits encoded in the mitochondrial DNA (Cox1, Cox2, and Cox3). Using a transport-dead Aac2 mutant, we previously demonstrated that AAC transport

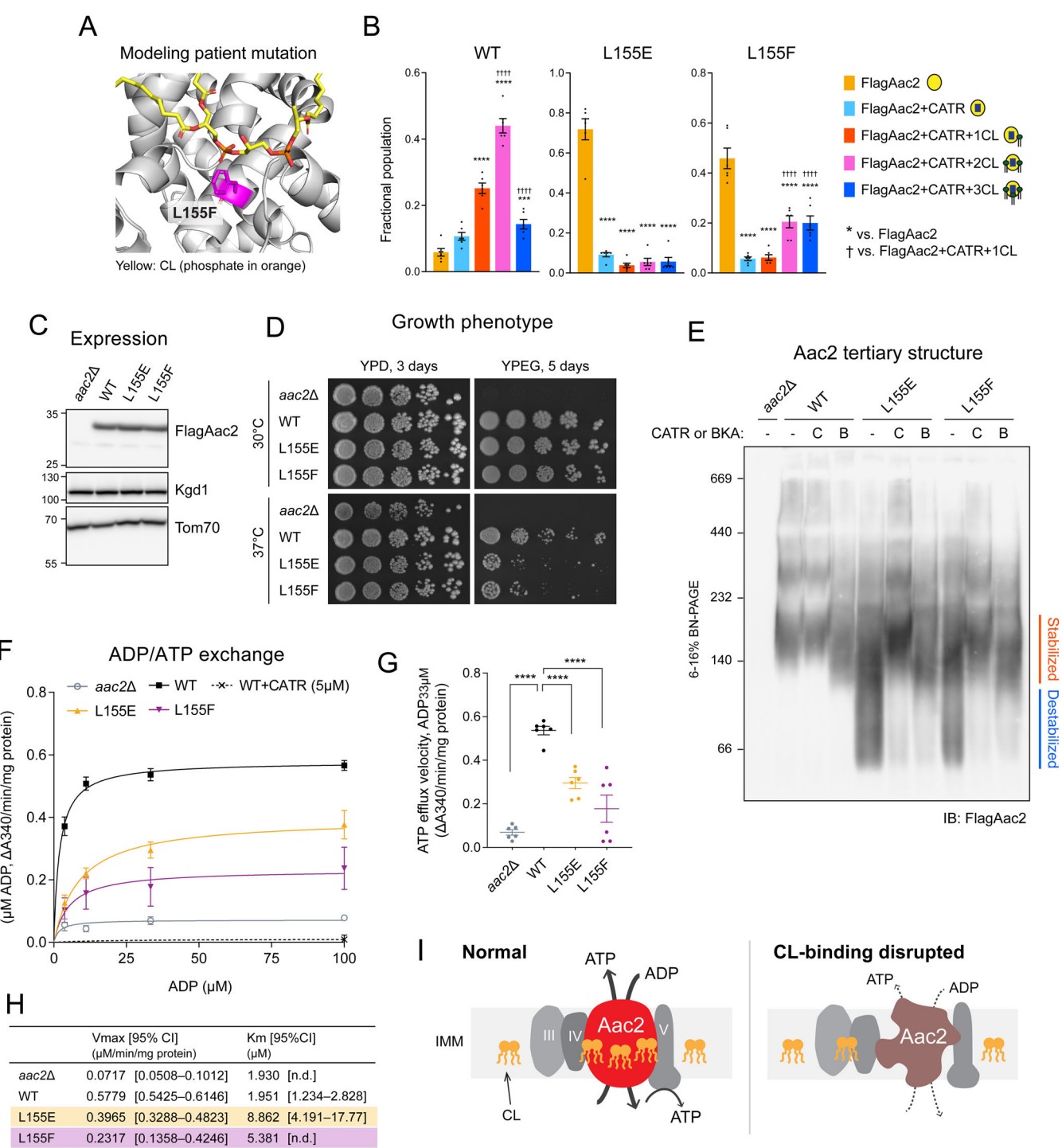

supports complex IV expression via modulating translation of its mitochondrial DNA-encoded subunits (Ogunbona et al, 2018). In this context, we noted that the mitochondrial steady-state expression of Cox1 and Cox3 were modestly reduced for those CL-binding Aac2 mutants whose transport activity was impaired the most (Figs. 4 and EV3). We found that the activity of complex III was mostly unaffected in Aac2 CL-binding mutants, with the exception of pocket 2 mutants, which showed slightly decreased activity (Fig. EV4A). In contrast, complex IV activity was impaired

and correlated with the mutants' transport ability (Fig. EV4B). Consistent with our previous finding (Ogunbona et al, 2018), these results indicate that AAC transport dysfunction diminishes the expression and activity of respiratory complexes, particularly complex IV; this likely accounts for the reduced CCCP-induced respiration for CL-binding mutants with AAC transport dysfunction (Fig. EV2A,B). The results reinforce our conclusion that interactions between CL and Aac2 are crucial for AAC transport function. The mutants still maintained the expression of nuclear

**Figure 5.   Disturbed Aac2–CL interaction is a pathological mechanism.**

(A) Mitochondrial myopathy patient mutation L155F was introduced into yeast Aac2 as Fig. 1A. (B) Native MS analysis obtained the fractional population of WT and indicated FlagAac2 mutants associated with CATR and one to three CL molecules. Mean with SEM ($n = 6$: 2 biological replicates with 3 technical replicates). Significant differences were obtained by one-way ANOVA with Tukey's multiple comparisons test (* vs. FlagAac2; † vs. FlagAac2 + CATR + 1CL); ***/††† $p < 0.001$, ****/†††† $p < 0.0001$. (C) Expression of WT and mutant Aac2 was detected in whole cell extract by immunoblot. Kgd1 and Tom70 served as loading controls ($n = 4$, biological replicates). (D) Growth phenotype of Aac2 CL-binding mutants. Serial dilutions of indicated cells were spotted onto fermentable (YPD) and respiratory (YPEG) media and incubated at 30 or 37 °C for 3–5 days ($n = 4$, biological replicates). (E) Aac2 tertiary structure: 100 μg of mitochondria from WT and Aac2-binding mutants were mock-treated or instead incubated with either 40 μM CATR (C) or 10 μM BKA (B) and then solubilized with 1.5% (w/v) digitonin. The extracts were resolved by 6 to 16% blue native-PAGE and immunoblotted for FlagAac2 ($n = 6$, biological replicates). (F–H) ADP/ATP exchange: The efflux of matrix ATP was detected with isolated mitochondria as NADPH formation (A340; absorbance at 340 nm) occurring coupled with in vitro glycolysis reaction as in Fig. 4. The measurement was performed in the presence of 5 mM malate and 5 mM pyruvate ($n = 6$, biological replicates). (F) The linear part of the initial velocity following the addition of ADP at indicated concentrations was plotted (mean with SEM). Curve fitting was performed by nonlinear regression. (G) The linear part of velocity when 33 μM ADP was added shown as scatter plots (mean with SEM). (H) Fitted $K_m$ and $V_{max}$ values were obtained by the Michaelis–Menten equation from the replicated experiments (mean). Significant differences were obtained by one-way ANOVA with Tukey's multiple comparisons test (vs. WT); ****$p < 0.0001$. (I) Predicted roles of the buried CLs within Aac2. If CLs are dissociated, the Aac2 tertiary structure is destabilized and ADP/ATP transport is compromised, which disrupts energy production via OXPHOS. In (C–E), representative images from the indicated replicates are shown. Source data are available online for this figure.

genome-encoded subunits of respiratory complexes (complex III, complex IV, and complex V) and the mitochondrial DNA-encoded subunit of complex V, Atp6 (Fig. EV3), as well as complex V activity (Fig. EV4C).

Since the ability of Aac2 to interact with respiratory complex subunits is sensitive to its transport-related conformation which is in turn influenced by CL (Senoo et al, 2020), we biochemically interrogated the Aac2-respiratory supercomplex (RSC) interaction for each CL-binding Aac2 mutant. As shown in Appendix fig. S6, blue native-PAGE results illustrated the higher-order protein assembly, including Aac2 and RSC. In the absence of CL (*crd1Δ*), the WT Aac2-RSC interaction was significantly destabilized (Appendix Fig. S6A,B), as expected (Claypool et al, 2008; Senoo et al, 2020). While less noticeable than WT Aac2 in *crd1Δ*, the assembly of the Aac2 CL-binding mutants with RSC was compromised to varying extents (Appendix Fig. S6A,B). Quantification indicated that the abundance of Aac2-III$_2$IV$_2$ was reduced in I51E (pocket 1), all three pocket 2 mutants, three pocket 3 mutants (R191D, M255E, and G267E), and the double mutant G69D/G267E (Appendix Fig. S6B). This suggests that disturbed CL-Aac2 interactions have a mild but measurable impact on Aac2-RSC interaction. While the abundance of RSCs (III$_2$IV$_2$ and III$_2$IV$_1$) was retained in all mutants (Appendix Fig. S6C,D), the fraction of III$_2$IV$_2$ was lowered (Appendix Fig. S6E). To further investigate direct interactions between Aac2 and RSC subunits, we performed Flag co-immunoprecipitation and used CATR as a folding stabilizer prior to the detergent solubilization. The pocket 2 mutant, N90E, barely interacted with the RSC subunits tested (Appendix Fig. S7). Impaired protein–protein interactions were also observed for L155E and G172E (pocket 2) and R191D (pocket 3) (Appendix Fig. S7C). The protein assembly defects consistently observed in the blue native-PAGE and co-immunoprecipitation results may stem from their severely weakened conformational stability and/or the relatively strong transport defect of these mutants. In fact, mutants with the most compromised transport co-purified the least amount of complex III and IV subunits (Appendix Fig. S7).

## Disturbed CL-ANT1 binding underpins an uncharacterized pathogenic mutation

More than half of the CL-binding mutant residues that we targeted in yeast Aac2 are conserved in mammalian ANTs (Fig. EV5). Among the conserved residues, the uncharacterized L141F

missense mutation in human ANT1—corresponding to yeast Aac2 L155 in pocket 2—was recently identified in a patient suffering from mitochondrial myopathy, exercise intolerance, hyperlactatemia, and cardiomyopathy (Tosserams et al, 2018). Since we observed that all yeast Aac2 mutants had destabilized tertiary structure (Fig. 3) and that L155 in pocket 2 was one of the crucial residues for Aac2 transport activity (Fig. 4), we hypothesized that the pathological mechanism behind this patient mutation may stem from disturbed lipid–protein interaction-based AAC/ANT dysfunction. To begin to characterize the patient mutation, we generated yeast Aac2 L155F. Due to its substantial girth, the insertion of Phe in this pocket was predicted to extrude CL due to steric hindrance (Fig. 5A). This prediction was supported as apo Aac2 was the most dominant population for Aac2 L155F (Fig. 5B), consistent with the corresponding Glu mutant (Figs. 2D, 5B). Since the 2- and 3-CL bound forms were comparatively retained (Fig. 5B), it is likely that the pathogenic Phe residue has a milder impact on pocket 2's CL association than Glu or Asp residues have. Similar to other CL-binding mutants described earlier, there was no obvious charge state shift for Aac2 L155F (Appendix Fig. S8), indicating this mutant is properly folded. Aac2 L155F was expressed at a similar level to WT (Fig. 5C) and maintained growth capacity in fermentable and respiratory media like L155E (Fig. 5D). The tertiary structure of Aac2 L155F was destabilized on blue native-PAGE and rescued by pretreatment with CATR or BKA (Fig. 5E). Moreover, Aac2 L155F had significantly reduced ADP/ATP exchange capacity (Fig. 5F–H). Similar to the other CL-binding pocket mutants (Fig. EV4), complex IV activity trended to be decreased for Aac2 L155F, whereas the activities of complexes III and V were maintained (Appendix Fig. S9). These results demonstrate that the patient mutant failed to support Aac2 tertiary structure and transport activity likely due to a perturbed lipid–protein interaction.

We sought to extend our investigation of the patient mutation into human ANT1 as expressed in a human cell model. To this end, we took advantage of ant$^{null}$ 293 cells in which all three major ANT isoforms were knocked out (Appendix Fig. S10) and re-introduced WT and the mutant ANT1 alleles (L141E and L141F). As in yeast Aac2, the introduced L141E and L141F mutations in human ANT1 were predicted to dismiss CL from the peptide backbone (Fig. 6A): L141E could repulse the CL phosphate and L141F sterically block CL engagement. WT and mutant ANT1 were equally expressed

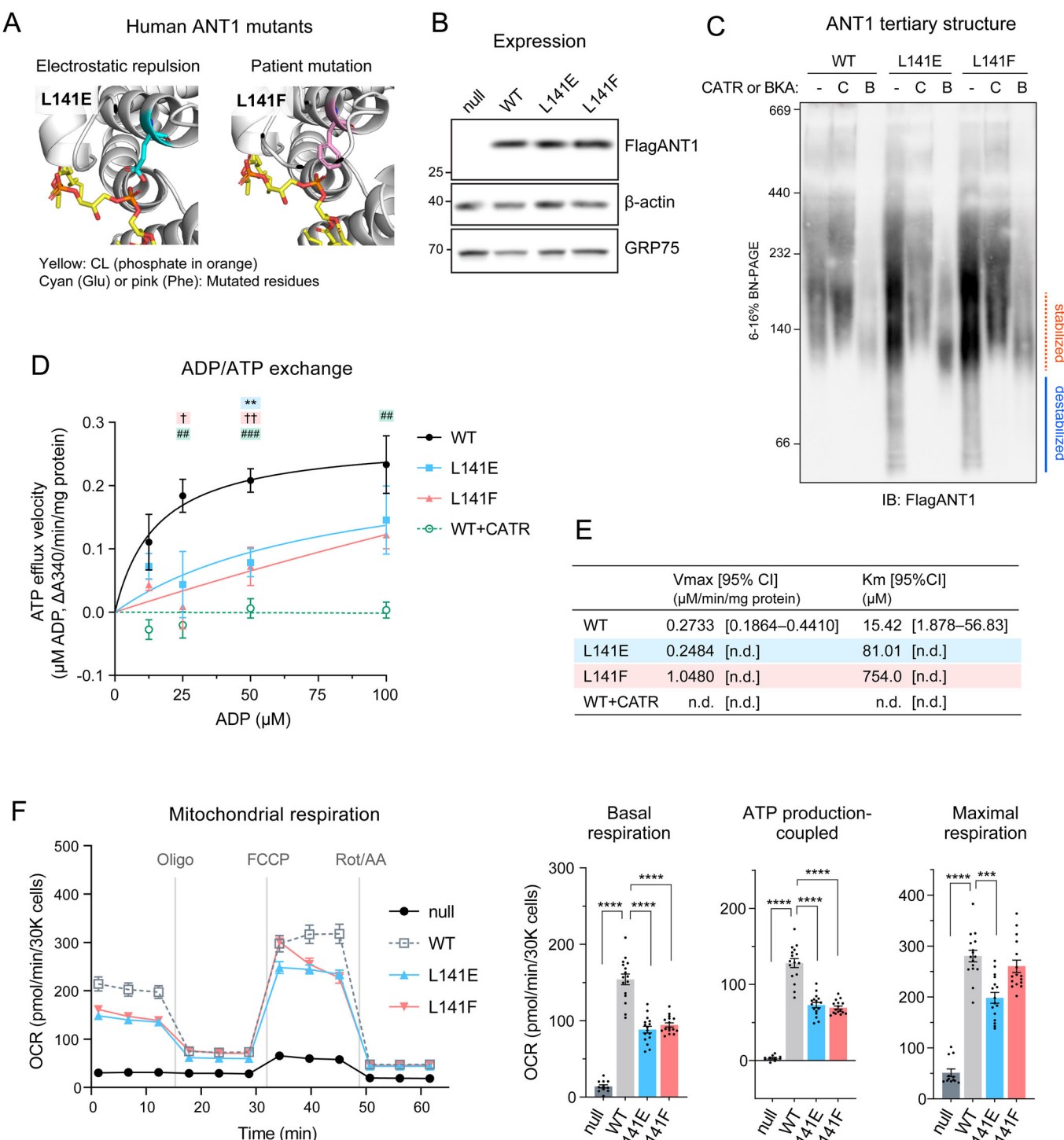

(Fig. 6B). The steady-state expression of select complex I (NDUFB6) and IV (COX4) subunits was decreased in ant$^{null}$ and L141F cells but preserved in L141E cells (Appendix Fig. S11). As determined by blue native-PAGE, WT ANT1 migrated around 140 kDa, which based on the yeast model, is predicted to reflect the stabilized tertiary structure (Fig. 6C). Pretreatments with CATR and BKA stabilized WT ANT1 at slightly different sizes (Fig. 6C); this migration pattern was identical to yeast Aac2 and likely reflects

the conformational status of ANT1 (c-state and m-state) (Senoo et al, 2020). Intriguingly, ANT1 L141E and L141F migrated as destabilized smears (Fig. 6C), suggesting that CL-binding to ANT1 in the vicinity of L141 (pocket 2) plays a significant role in stabilizing the carrier's tertiary structure. CATR or BKA pretreatment preserved the structure of L141E and L141F, indicating that the mutant forms of ANT1 were properly folded within the membrane pre-solubilization. The ADP/ATP exchange capacities

◄

**Figure 6. Human ANT1 L141F are structurally and functionally compromised.**

(A) Human ANT1 was modeled onto bovine ANT1 (PDB ID: 2C3E) using SWISS-MODEL. (B–F) Flag-ANT1 and the indicated ANT1 mutants were induced by 0.25 µg/ml doxycycline in ant$^{null}$ T-REx-293 cells with three ANT isoforms (ANT1, 2, and 3) knocked out. (B) Expression of ANT1 was detected in whole cell extracts by immunoblot. β-actin and GRP75 were loading controls ($n = 3$, biological replicates). (C) ANT1 tertiary structure: 80 µg of mitochondria from WT and ANT1 mutants were mock-treated or instead incubated with either 40 µM CATR or 10 µM BKA and then solubilized with 1.5% (w/v) digitonin. The extracts were resolved by 6 to 16% blue native-PAGE and immunoblotted for Flag-ANT1 ($n = 3$, biological replicates). (D, E) ADP/ATP exchange: The efflux of matrix ATP was detected with isolated mitochondria as NADPH formation (A340; absorbance at 340 nm) as in Fig. 4. The measurement was performed in the presence of 5 mM malate and 5 mM pyruvate. 5 µM CATR was added to WT mitochondria prior to stimulating the efflux ($n = 3$, biological replicates). (D) The initial velocity following the addition of ADP at indicated concentrations was plotted (mean with SEM). Curve fitting was performed by nonlinear regression. Significant differences obtained by one-way ANOVA with Dunnett's multiple comparisons test are shown as *, L141E; †, L141F; #, WT + CATR (vs. WT). *$p < 0.05$, **$p < 0.01$, ***$p < 0.001$. (E) Fitted $K_m$ and $V_{max}$ values from the Michaelis–Menten equation (mean). (F) Cellular oxygen consumption rate (OCR) was measured using a Seahorse XF96e FluxAnalyzer with the Mito Stress Test kit under indicated conditions. Basal and maximal OCR were obtained under glucose stimulation after FCCP treatment to uncouple mitochondria. ATP production-coupled respiration is defined as basal OCR subtracted by post-oligomycin OCR. Significant differences were determined by one-way ANOVA with Tukey's multiple comparisons test (vs. WT), ***$p < 0.001$ ****$p < 0.0001$. Means with SEM ($n = 16$, 3 biological replicates with 4–8 technical replicates). In (B) and (C), representative images from the indicated replicates are shown. Source data are available online for this figure.

of L141E and L141F were significantly impaired (Fig. 6D,E). Oxygen consumption measured in cells expressing mutant ANT1 showed compromised basal and ATP production-coupled respiration for L141E and L141F and reduced maximal respiration for L141E-expressing cells (Fig. 6F).

To investigate the impact of L141 mutations on the molecular structure and CL interactions of ANT1, all-atom MD simulations of WT and mutant human ANT1 were performed in a CL-containing lipid bilayer (Appendix Fig. S12). Time-averaged probability density maps of CL positions relative to ANT1 were generated for the "CL prebound" (Fig. 7A) and "CL unbound" (Fig. 7B) simulations. The "CL prebound" and "CL unbound" simulations are defined by the presence (prebound) or absence (unbound) of a CL lipid in the vicinity of binding pocket 2 prior to the initial state of the simulations (See Methods and figs. S11B,C for further details). For the prebound simulations for WT and mutant ANT1, we observed a high density of CL (Fig. 7A) in the three known CL-specific binding pockets (Hedger et al, 2016; Corey et al, 2019; Mao et al, 2021; Duncan et al, 2018). In the prebound simulations, we did not observe a significant alteration to the CL densities around pocket 2. This may suggest longer time scales are required to capture the complete binding and unbinding processes of CL from the binding pockets. Another factor, in the case of the L141E mutant, is that the expected electrostatic repulsion between CL and the Glu residue is mitigated by the formation of a salt bridge interaction with the proximal R152 residue.

As an alternative to extending the simulation time, we performed a set of simulations in which the CL lipids were removed from pocket 2 (unbound). From these simulations, we observed for WT ANT1 a density map consistent with the prebound simulations showing a high density of CL located in the three known binding sites (Hedger et al, 2016; Corey et al, 2019; Mao et al, 2021; Duncan et al, 2018). In contrast, the mutant ANT1 systems displayed low CL density proximal to pocket 2 (Fig. 7B). The 2D lipid density profiles are supported by 2D radial distribution functions (g(r)), which represent the probability of locating the central glycerol of CL at varying radial distances from the Cα atom of residue 141. In the case of prebound simulations (Fig. 7C), both WT and mutant proteins had the highest probability of finding CL lipids to be at ~0.90 nm from residue 141, which is consistent with crystallographically determined distances (PDB ID 1OKC: $r_{141-CL} = 0.94$ nm; 2C3E: $r_{141-CL} = 0.96$ nm). In the case of the unbound simulations (Fig. 7D), a considerably higher

probability of CL lipids proximal to residue 141 was observed for WT compared to L141F or L141E mutant systems. Both WT and L141F displayed a plateau region centered around 1.3 nm, which may represent an approach state prior to CL accommodation in pocket 2. The bulky Phe residue of L141F may impede the accommodation of CL into pocket 2 and the lack of the approach plateau in L141E is likely due to the electrostatic repulsion between Glu and CL. Unlike the prebound simulations, a salt bridge between E141 and R152 is not observed for the L141E system in the unbound simulations. It is possible that the presence of CL in pocket 2 in the prebound simulations drove the formation of the E141-R152 salt bridge, and when CL is removed from the pocket (unbound), the formation of the salt bridge is less energetically favorable.

Protein structure and dynamics during the simulations were examined, and ANT1 was more stable in the CL prebound simulations than the CL unbound simulations, for WT and mutants (Appendix Fig. S13A,B). To examine the dynamics of residue 141 and the stability of pocket 2 residues, the distance between the Cα atom of residue 141 with that of the other pocket 2 binding residues was calculated for the prebound simulations (Appendix Fig. S13C). In the WT system, the arrangement of pocket 2 residues relative to L141 was highly stable. On the contrary, neighboring amino acids of residue 141, such as residues 152 and 155–158, displayed fluctuating distances, especially His155 in the L141F mutant system. The observed shorter distances between H155 and F141 in L141F indicate instances of $\pi - \pi$ stacking interactions. The L141E mutant system showed comparatively less variable distances between residue 141 and all other considered pocket 2 binding residues than the L141F mutant. This is potentially due to the observed salt bridge interaction between residue E141 and R152. Overall, the simulations support that mutations of L141 destabilize the CL binding environment and cause higher fluctuations in the residues presented between helices 3–4 (matrix-oriented loop). However, the mutations at pocket 2 do not appear to cause large-scale destabilization to the overall protein fold, as shown by RMSD analysis (Appendix fig. S13A) or by alpha-helical content (Appendix Fig. S14A). Furthermore, the local disruption caused by L141 mutations at pocket 2 does not appear to have significant structural effects on pocket 1 or 3, as analyzed by local RMSD calculations (Appendix Fig. S14C–F). In unbound simulations, the L141E mutant does display some larger structural fluctuations at both pocket 1 and pocket 3, compared to other systems, but the

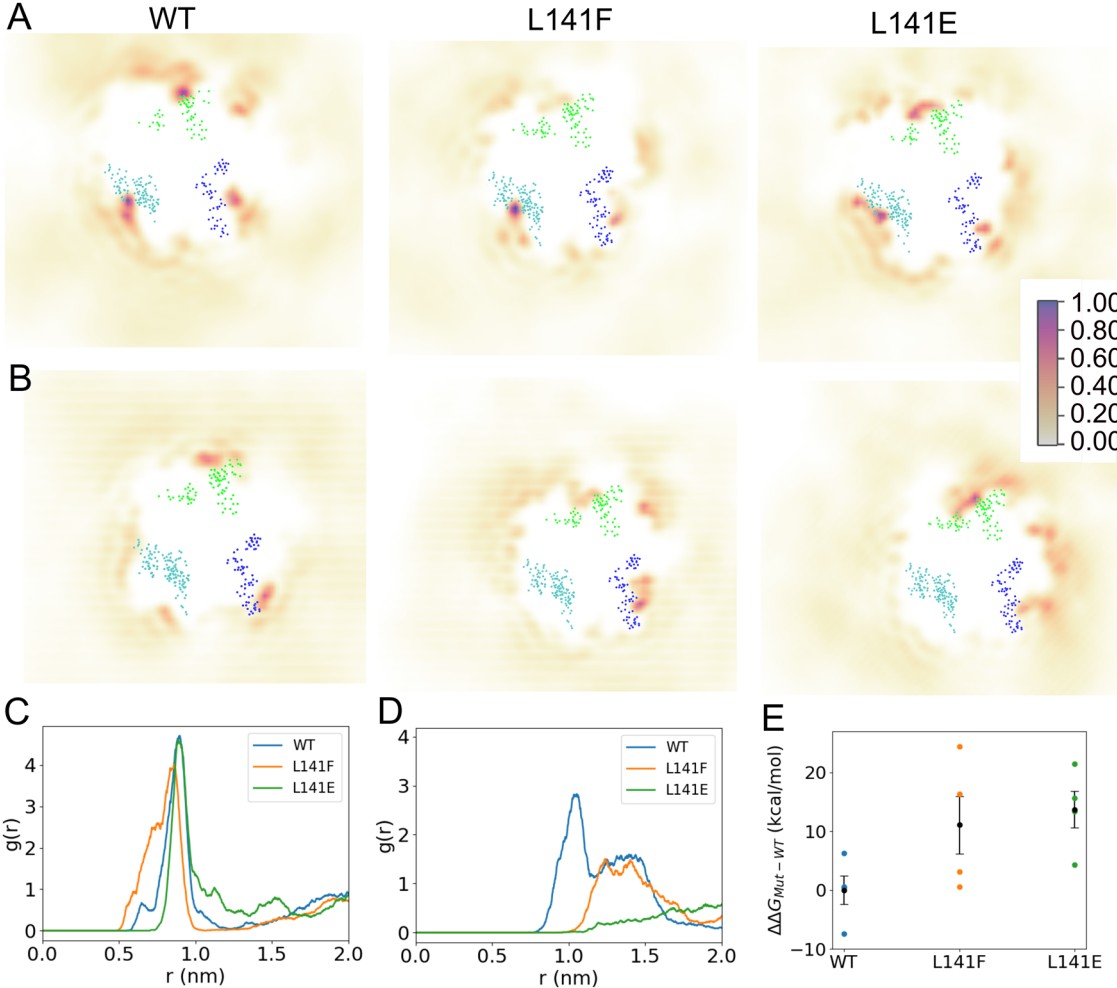

**Figure 7.  MD simulations predict reduced CL-ANT1 affinity at pocket 2 for L141 mutants.**

(A, B) Time-averaged 2D density maps of CL lipids for the equilibrium prebound (A) and unbound (B) simulations. The CL binding pockets are defined by the blue (pocket 1), cyan (pocket 2), and green (pocket 3) dots, which represent the amino acids that comprise each binding site. The density surfaces were calculated using vmd volmap with 1 Å resolution. The occupancy scale bar on the right applies to all images. (C, D) Radial distribution function profile for the identification of CL lipids around residue 141 for prebound (C) and unbound (D) simulation. (E) The estimated relative free energy required to decouple CL from binding pocket 2 of ANT1 for WT and mutants L141F and L141E. Using the Wilcoxson rank-sum test between WT and L141E indicates statistical significance ($p < 0.05$), while the difference between WT and L141F is not significant. Mean with SEM ($n = 4$). Note: Binding site residues considered in the present study are based on the protein-CL lipid interactions and are the selected amino acid residues within 8 Å of the bound CL molecules: **Pocket 1:** 36, 53, 54, 55, 271, 272, 273, 274, 275, and 276; **Pocket 2:** 71, 72, 73, 74, 75, 141, 152, 155, 156, 157, and 158; **Pocket 3:** 251, 252, 253, 254, 255, 174, 175, 176, 177, and 178, respectively. Source data are available online for this figure.

increased RMSD is not sustained, and at the end of the simulations the RMSD value returns to a value consistent with WT and L141F simulations.

Assimilating the prebound (Fig. 7A,C) and unbound (Fig. 7B,D) simulation results, the CL interaction at pocket 2 is stable in both WT and mutants, but the mutations may create a barrier for CL to access the binding site, resulting in the reduced pocket 2 density in the mutants. Note that the equilibrium simulations are not sufficiently long to capture unbinding/rebinding events to accurately calculate populations (i.e., free energy differences), and therefore, we used a free energy perturbation (FEP) approach to estimate the difference in CL binding free energies between WT and mutants (Appendix Fig. S15). The relative free energy required to decouple a CL molecule from pocket 2 of ANT1 WT, L141F, and L141E systems are shown in Fig. 7E. From the calculated $\Delta\Delta G$

values, WT ANT1 showed ~12 kcal/mol more favorable binding energy than the two studied mutant systems. In toto, these combined experimental and simulation results demonstrate that CL-binding at pocket 2 of human ANT1 is both structurally and functionally important and that the mitochondrial dysfunction associated with the previously uncharacterized L141F pathogenic mutation (Tosserams et al, 2018) derives from a perturbed CL-ANT1 interaction.

## The ability of CL to stabilize Aac2 tertiary structure is specific

Finally, to clarify the causal relationship between disturbed CL binding and AAC dysfunction, we investigated additional yeast Aac2 mutations with documented transport defects (Ogunbona

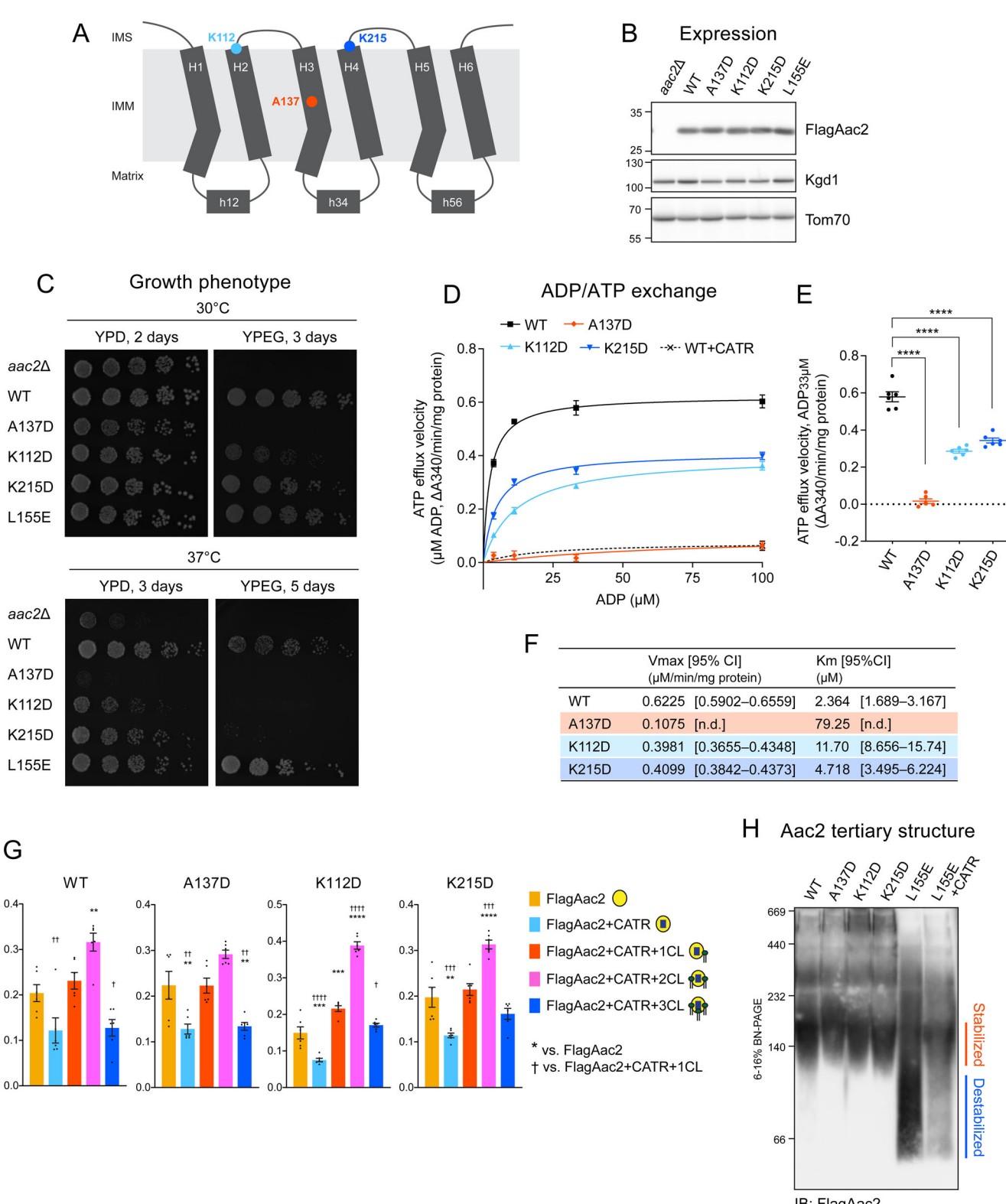

et al, 2018; Palmieri et al, 2005; Ruprecht et al, 2014, 2019; King et al, 2016). Given their location within Aac2 (Fig. 8A), these additional transport-defective mutants are not expected to impact CL bindings which occur at the matrix side (Fig. 2B). We focused on three residues and replaced them with Asp: a previously

characterized transport-null pathogenic mutation A137D located in the transmembrane region (Ogunbona et al, 2018; Palmieri et al, 2005) and two transport hypomorphic mutations that disrupt a salt-bridge network on the cytoplasmic side, K112D and K215D (Ruprecht et al, 2014, 2019; King et al, 2016) (Fig. 8B). As expected,

**Figure 8.  CL binding is preserved in additional transport-defective Aac2 mutants.**

(**A**) Schematic representation showing the positions of non-CL binding related mutations in Aac2. (**B**) Expression of WT and indicated FlagAac2 mutants. Kgd1 and Tom70 served as loading controls ($n = 3$, biological replicates). (**C**) Growth phenotype of Aac2 CL-binding mutants. Serial dilutions of indicated cells were spotted onto fermentable (YPD) and respiratory (YPEG) media and incubated at 30 or 37 °C for 3–5 days ($n = 6$, biological replicates). (**D–F**) ADP/ATP exchange: The efflux of matrix ATP was detected as in Fig. 4. The measurement was performed in the presence of 5 mM malate and 5 mM pyruvate ($n = 6$, biological replicates). (**D**) The linear part of the initial velocity following the addition of ADP at indicated concentrations was plotted (mean with SEM). Curve fitting was performed by nonlinear regression. (**E**) The linear part of velocity when 33 µM ADP was added is shown as scatter plots (mean with SEM). (**F**) Fitted $K_m$ and $V_{max}$ values were obtained by the Michaelis–Menten equation from the replicated experiments (mean). Significant differences were obtained by one-way ANOVA with Tukey's multiple comparisons test (vs. WT); ****$p < 0.0001$. (**G**) Native MS analysis obtained the fractional population of WT and indicated FlagAac2 mutants associated with CATR and one to three CL molecules. Mean with SEM ($n = 6$: 2 biological replicates with 3 technical replicates). Significant differences were obtained by one-way ANOVA with Tukey's multiple comparisons test (* vs. FlagAac2; † vs. FlagAac2+CATR + 1CL); */† $p < 0.05$, **/†† $p < 0.01$, ***/††† $p < 0.001$, ****/†††† $p < 0.0001$. (**H**) Aac2 tertiary structure: 100 µg of mitochondria from WT and the indicated mutants were mock-treated or instead incubated with either 40 µM CATR and then solubilized with 1.5% (w/v) digitonin. The extracts were resolved by 6 to 16% blue native-PAGE and immunoblotted for FlagAac2 ($n = 4$, biological replicates). Source data are available online for this figure.

the A137D mutant failed to support respiratory growth (Fig. 8C) and was largely devoid of transport activity (Fig. 8D–F). In contrast, the two salt-bridge mutants retained some, albeit less than WT, respiratory growth (Fig. 8C) and had ~50% reduced transport capacity compared to WT (Fig. 8D–F). Native MS analysis showed that A137D, K112D, and K215D mutants quantitatively associated with 2- or 3- CLs the same as WT (Fig. 8G). This result indicates that these mutations had no impact on CL bindings, in turn demonstrating that CL binding disruption is not attributed to nonspecific or secondary effects of AAC transport dysfunction. Blue native-PAGE analysis illustrated that A137D, K112D, and K215D mutants migrated like WT, demonstrating that their tertiary structure is intact (Fig. 8H). Based on these results, we conclude that the ability to stabilize AAC/ANT tertiary structure is a specific function of three evolutionarily conserved CL binding events.

# Discussion

Previous structural studies identified three CL molecules tightly bound to AAC in an evolutionarily conserved manner (Ruprecht et al, 2014; Nury et al, 2005); however, the functional significance of these lipid–protein interactions has not been determined. In this study, we engineered yeast Aac2 mutants to disrupt CL binding; the dissociated lipid–protein interactions of the mutants were confirmed by native MS analysis (Fig. 2). The results provide experimental evidence that the CL molecules bound to Aac2 stabilize the carrier's tertiary structure and support its transport activity (Fig. 5I). Further, we identified that the residues mutated to disrupt CL-Aac2 interactions are conserved in mammalian ANTs, which includes a previously reported patient mutation (L141F in human ANT1) (Tosserams et al, 2018). When the patient mutation was modeled in yeast Aac2 (L155F) and human ANT1 (L141F), the pathogenic mutation compromised the carrier's structure and transport activity (Figs. 5,6) due to a disturbed lipid–protein interaction (Figs. 5B,7). This would represent, to the best of our knowledge, the first known disease-causing mutation which disrupts a structurally and functionally important lipid–protein interaction.

The patient mutant, yeast Aac2 L155F, somewhat retained its capacity to bind 2–3 CLs when assessed by our native MS system, even though apo Aac2 was the most dominant population for this mutant, similar to the negatively charged residue mutants (Figs. 2D, 5B). This suggests that the Phe residue has a relatively mild effect

on dissociating the targeted CL-Aac2 interaction compared to the electrostatic repulsion between CL phosphates and introduced negatively charged amino acids in the Glu or Asp mutants. Our MD simulations further clarified the CL-binding status of the patient mutant ANT1 L141F (Fig. 7). The unbound simulation indicated that L141F failed to accommodate CL in pocket 2 but retained a CL approaching event, in contrast to L141E which lost both events (Fig. 7D). Nonetheless, the extent of structural and functional impairments was comparable between the Glu and Phe mutants in yeast Aac2 and human ANT1 (Figs. 5,6). These results imply that the disruption of this CL-Aac2 interaction, even if incomplete, causes serious impairment.

Our results indicate that the CL molecules in each pocket behave distinctly. Whereas the mutants for pocket 2 (N90E, L155E, G172E) and pocket 3 (R191D, M255E, G267E) had severely compromised transport as well as OXPHOS capacity, the pocket 1 mutants (I51E and G69D) largely retained these activities. The double mutant (G69D/G267E) for pockets 1 and 3 showed a stronger defect in transport capacity than the corresponding single mutants. Our native MS results indicated that the strength of CL bindings is different among the pockets: one of the pockets has a relatively weak affinity, and the other two pockets have a similar affinity (Appendix Fig. S2). In repeated experiments to obtain fractional populations using distinct preparations of mitochondria, we noted that the preferred interaction status for WT Aac2 varied slightly: CATR-bound Aac2 + 3 CLs was the highest in Fig. 2B whereas CATR-bound Aac2 + 2 CLs was the highest in Figs. 5B, 8D. While the basis for this difference is unclear, we note that even when Aac2 + 2 CLs was dominant, a significant amount of Aac2 + 3 CLs remained. In the net, these data indicate that WT Aac2 binds up to three CLs and also strongly suggest that one of the three CL interactions engages Aac2 with a lower affinity than the other two CLs. Although we expected that fractional population results of the mutants from the native MS analyses might give us insight into the CL-binding affinity of each pocket, the broad impact on CL binding of most mutants made this difficult. Recent MD simulations of fungal AAC in the m-state demonstrated that CLs in pockets 2 and 3 (CDL800 and CDL802) are constantly bound, while CL in pocket 1 (CDL801) is more labile (Montalvo-Acosta et al, 2021). Supporting our MSMS results, these simulation results suggest that the functional difference across pockets that we observed may correlate with their binding affinity for CLs. Another MD simulation of bovine ANT1 in the c-state indicated that pocket 2 is densely associated with CL, and this CL contributes to stabilizing

the carrier (Yi et al, 2022). Collectively, pockets 2 and 3 appear to have preeminent roles, while the labile CL molecule in pocket 1 would have a more nuanced and/or specific Aac2-related role. Although we performed MD simulations only on the pocket 2 mutant of human ANT1, the simulations will be extended to the other CL-binding pocket mutants with physiological CL amount and acyl chain profile (e.g., representing the human heart), which would further clarify the binding affinity across pockets and provide insight into the physiological significance of these specific lipid interaction events.

We consider that the impact of the inserted CL-binding mutations on protein structure is minimal based on the following four observations: (1) As biochemically tested, the topology of the CL-binding mutants in the IMM was identical to WT Aac2 (Fig. 1E); (2) As assessed by blue native-PAGE, the addition of CATR and BKA to pre-digitonin solubilized mitochondria rescued the tertiary structure of the CL-binding mutants (Fig. 3); (3) the charge state shift for each CL-binding mutant was minimal in the native MS analyses (Appendix Fig. S4); and (4) RMSD analysis and alpha-helical content did not indicate significant effects of pocket 2 mutations in human ANT1 on overall protein fold and pockets 1 and 3 structures (Appendix Figs. S13, 14).

Worth considering is that some of the inserted mutations are likely to exert non-CL binding related effects. For instance, N90E and R191D are located in the vicinity of several residues recently shown to be critical for substrate binding related to the AAC transport cycle (Mavridou et al, 2022). It would seem plausible that the negatively charged residues we inserted might affect substrate binding itself. Indeed, these two mutants had severe respiratory growth defects (Fig. 1C). In addition, the N90E mutant had a weakened association with respiratory complex subunits compared to the other mutants tested (Appendix Fig. S7). Still, CL binding was disrupted for both the N90E and R191D Aac2 mutants (Fig. 2D). As such, their defects may be attributed to a combination of disturbed lipid–protein interaction and impaired substrate binding, and both may be affected in a distinct and/or synergized manner.

Moving forward, there are several important avenues to pursue based on our present findings. While the current study focuses on the roles of CL in AAC/ANT's ADP/ATP exchange function, it is worth noting mammalian ANTs have recently been shown to also mediate proton translocation (Bertholet et al, 2019). Given that there appears to be a reciprocal relationship between ADP/ATP exchange and proton leak (Bertholet et al, 2019), it will be important in the future to determine the proton transport capacity of human CL-binding ANT1 mutants. Another avenue of future investigation should focus on if and how different CL species differ in their ability to support AAC/ANT structure and activity. A classic study from the Klingenberg Lab focused on the association between CL and bovine ANT1 concluded that the acyl chain composition of CL, including its saturation status, does not impact its ability to strongly engage the carrier (Schlame et al, 1991). In support of this conclusion, we previously showed that yeast Aac2 assembly and function is the same in mitochondria with unremodeled CL (lacking the CL deacylase, Cld1) as with fully remodeled CL (wildtype; (Baile et al, 2014)). Interestingly, we did note that in the absence of the monolyso-CL transacylase, TAFAZZIN (Schlame and Xu, 2020), the amount of Aac2 associated with respiratory supercomplexes is decreased (Baile et al, 2014). As *TAFAZZIN* is mutated in Barth syndrome (Barth

et al, 2004; Clarke et al, 2013), an X-linked cardio- and skeletal myopathy, this implicates potential defects in ANT assembly and function in this disease. Still, the acyl chain complexity of CL in mammals relative to laboratory yeast is vast and includes tissue-specific differences in the final predominant molecular forms of CL and the incorporation of polyunsaturated fatty acids (PUFA) of various lengths (Oemer et al, 2020). Incorporation of PUFAs into CL are influenced by the activities of delta 6 desaturase, the rate-limiting enzyme in generating long-chain PUFAs (Mulligan et al, 2014), and ALCAT1, an endoplasmic reticulum-resident mono-lyso-CL acyltransferase (Li et al, 2010). PUFA-containing CL species are intrinsically more susceptible to oxidative damage and tied to impaired metabolism (Li et al, 2010) and aging (Mulligan et al, 2014). To our knowledge, the ability of long-chain PUFA-containing CLs to fully support the structure and function of mammalian ANT, and whether these roles are altered in a meaningful way upon CL peroxidation (Kagan et al, 2023) have not been tested, and such insight could provide a more detailed understanding of how perturbed CL metabolism contributes to disease pathogenesis.

Most of the carriers belonging to the SLC25 family are embedded in the IMM like AAC. Their topology and transport-related conformations have been predicted to be very similar to AAC (Ruprecht and Kunji, 2019). Given that CL is enriched in the IMM, CL may be widely involved in supporting the structure and transport activity of the extended SLC25 family. In fact, a very recent study indicated that CL binding is involved in the activity of SLC25A51, also known as the mitochondrial NAD$^+$ carrier (Goyal et al, 2023). Our findings highlight the structural and functional significance of tightly-bound CL in yeast Aac2 and human ANT1. This serves as a paradigm for the important roles exerted by specific lipid–protein interactions in mitochondria and provides a clinically relevant example of conserved lipid–protein interactions in a membrane-integrated carrier protein.

## Limitations of this study

In mitochondria, AAC/ANT-mediated exchange of ADP and ATP, respiratory complex function, and ATP synthesis are intimately integrated in the IMM. Due to this necessary functional coupling, it is challenging to isolate mutant AAC/ANT function from unexpected impacts on respiratory complexes and ATP synthase using intact and, thus, active mitochondria. As described earlier, AAC transport function is essential for complex IV expression and activity (Ogunbona et al, 2018), and AAC transport-related conformational changes affect protein–protein interactions between Aac2 and respiratory complexes (Senoo et al, 2020). Indeed, we observed measurable respiratory complex defects in Aac2 CL-binding mutants that correlated with the severity of their transport dysfunction, based on the results of complex IV subunit expression (Fig. EV3) and activity (Fig. EV4), Aac2-RSC interactions (Appendix Figs. S6, 7), and CCCP-induced respiration rates (i.e., uncoupled respiration from ADP phosphorylation) (Fig. EV2). In accounting for ATP synthesis, a technical limitation is that commonly used methods using mitochondria rely heavily on an exchange of ADP and ATP; for example, the reaction is initiated by adding exogenous ADP to mitochondria and tracked by measuring the produced ATP by luminescence detection (Lanza and Nair, 2009). To isolate complex V function independent of the IMM, we

performed complex V in-gel activity assays (Fig. EV4C). While our data indicate that complex V activity of Aac2 CL-binding mutants was intact and, therefore, unlikely to contribute to the decreases observed in AAC transport activity, this assay detects ATP hydrolysis and not synthesis (Zerbetto et al, 1997). Ultimately, in vitro reconstitution into liposomes in the presence or absence of pharmacologically generated membrane potentials are needed to clarify the transport activity intrinsic to each AAC/ANT CL-binding mutant in isolation, an important future goal.

# Methods

## Yeast strains and growth conditions

All yeast strains used were derived from *S. cerevisiae* parental strain GA74-1A (*MATa, his3-11,15, leu2, ura3, trp1, ade8, rho+, and mit+*). *aac2Δ* were generated by replacing the entire open reading frame of AAC2 with markers HIS3MX6, as previously described (Claypool et al, 2008) via polymerase chain reaction (PCR)–mediated gene replacement (Wach et al, 1994).

Single point mutations into the Aac2 open reading frame were introduced by PCR-based overlap extension using primers containing each mutant sequence. To place the Flag tag onto the N-terminus of Aac2 still downstream of the promoter, PCR-mediated overlap extension was performed against either WT Aac2 or each mutant open reading frame using primers containing Flag sequence (DYKDDDDK). All were cloned into pRS305. Primers used to generate the constructs are listed in Appendix Table S1. The sequences of every construct were verified by Sanger DNA sequencing. The pRS305 constructs were linearized and integrated into the *LEU* locus in the *aac2Δ* background. Clones were selected on synthetic dropout medium (0.17% (w/v) yeast nitrogen base, 0.5% (w/v) ammonium sulfate, 0.2% (w/v) dropout mixture synthetic-leu, 2% (w/v) dextrose) and verified by immunoblot.

*crd1Δ* strain was generated by a homology-integrated CRISPR-Cas system as previously described (Bao et al, 2015; Ogunbona et al, 2017; Calzada et al, 2019). CRISPR-Cas9 gene block was designed to target *CRD1* and cloned into pCRCT plasmid (Bao et al, 2015). pCRCT was a gift from Huimin Zhao (Addgene plasmid #60621; http://n2t.net/addgene:60621; RRID:Addgene_60621). The sequence of *CRD1* gene block is: 5'-CTTTGGTCTCACCAAAACG ATGTCCTAGAAAGAGGAGATGAATTTTTAGAAGCCTATCC CAGAAGAAGGCAGATTTCTTTCATACAGTACATCATTACT GACCTTCGGTGTATCAAAAGTATGAAAGAAATCTCCAAAG TTTTAGAGAGAGACCTTTC-3'.

To map the epitope(s) recognized by the in-house generated ANT2 antiserum (Acoba et al, 2021), we subcloned WT ANT3, ANT2, or the indicated ANT2 epitope mutants in which ANT2-specific residues were replaced by their ANT3 counterparts by overlap extension PCR, into pRS315 (Fig. S10A,B). Of note, functional expression of human ANT isoforms in yeast requires the substitution of a stretch of mammalian N-terminal amino acids for those from yeast Aac2 (Hamazaki et al, 2011), which allows their detection with the Aac2-specific monoclonal antibody, 6H8 (Panneels et al, 2003). All constructs were heterologously expressed in *aac2Δ* yeast and grown in synthetic dropout media (0.17% yeast nitrogen base, 0.5% ammonium sulfate, 0.2% dropout mix synthetic –leu) supplemented with 2% dextrose.

Yeast were grown in nutrient-rich YP [1% (w/v) yeast extract and 2% (w/v) tryptone] media that contained 2% (w/v) dextrose (YPD). For growth analysis on solid plates, yeast cells were grown in YPD before spotting onto YPD or YPEG [1% (w/v) yeast extract, 2% (w/v) tryptone, 1% (v/v) ethanol, and 3% (v/v) glycerol] plates [2% (w/v) Bacto agar]. For mitochondrial isolation, cells were grown in YP-Sucrose [1% (w/v) yeast extract and 2% (w/v) tryptone, 2% (w/v) sucrose].

## Human cell lines and culture conditions

T-REx™ 293 cells (Invitrogen R71007) were grown on a plate coated by Cultrex Rat Collagen I (50 µg/ml, R&D systems 3440-100-01) at 37 °C in 5% $CO_2$ in high glucose-containing DMEM (Corning 10-013-CM) supplemented with 10% FBS (CPS Serum FBS-500), 2 mM L-glutamine (Gibco 25030081), 1 mM sodium pyruvate (Sigma S8636), 50 µg/ml uridine (Sigma U3003), and 15 µg/ml blasticidin (InvivoGen ant-bl-1). We have ruled out the presence of mycoplasma in the cells through routine testing.

ant[null] cells were established by knocking out ANT1, ANT2, and ANT3 in T-REx™ 293 cells. Individual gRNAs that targeted ANT1, ANT2, and ANT3 were separately cloned into pSpCas9(BB)-2A-Puro (PX459) V2.0, a gift from Feng Zhang (Addgene plasmid #62988; http://n2t.net/addgene:62988; RRID:Addgene_62988)(Ran et al, 2013). gRNA target sequences are 5'-GATGGGCGC-TACCGCCGTCT-3' for ANT1, 5'-GGGGAAGTAACGGAT-CACGT-3' for ANT2, and 5'-CGGCCGTGGCTCCGATCGAG-3' for ANT3, respectively. Transfections of the PX459 constructs were performed individually using FuGENE 6 Transfection Reagent (Promega E2691) according to the manufacturer's protocol. Following each transfection, cells were selected with puromycin (2 µg/ml) for 72 h, single clones were isolated by ring cloning, the absence of the protein expression was analyzed by immunoblotting, and the next transfection proceeded.

Flag-ANT1 was amplified by PCR using the previously published construct (Lu et al, 2017) as a template with primers designed to attach the Flag-tag onto the N-terminus of ANT1. Single point mutation into the ANT1 open reading frame was introduced by PCR-based overlap extension using primers containing each mutant sequence. Table S2 lists sequences of the primers used for these constructs. All were cloned into pcDNA™5/FRT/TO (Invitrogen V652020). ant[null] cells were co-transfected with pOG44 (Invitrogen V600520, expressing the Flp-recombinase) and the relevant pcDNA™5/FRT/TO plasmid at a ratio of 9:1 using FuGENE 6 Transfection Reagent. Transfected cells were selected using hygromycin B (Invitrogen) at 20 µg/ml for 2 days and then 40 µg/ml for 4 days. Individual clones isolated by single-cell seeding on a 96-well plate were subsequently expanded and screened by immunoblot. The cells introduced with Flag-ANT1 WT and mutants were maintained in glucose-based media [high glucose DMEM, 10% FBS, 2 mM L-glutamine, 1 mM sodium pyruvate, 50 µg/ml uridine, 15 µg/ml blasticidin] supplemented with 0.25 µg/ml doxycycline.

## Whole-cell protein extraction and immunoblotting

Protein extraction from yeast cells and immunoblotting were performed as detailed previously (Claypool et al, 2008; Calzada et al, 2019; Claypool et al, 2006). For human cells, proteins were

extracted from confluent six- or twelve-well tissue-culture dishes using RIPA lysis buffer [1% (v/v) Triton X-100, 20 mM HEPES-KOH, pH 7.4, 50 mM NaCl, 1 mM EDTA, 2.5 mM MgCl$_2$, 0.1% (w/v) SDS] spiked with 1 mM PMSF and quantified using the bicinchoninic acid assay (Pierce) as described in (Lu et al, 2016).

Antibodies used in this study are listed in Table S3. Custom-produced antibodies against yeast Tim54 were generated by Pacific Immunology (Ramona, CA) using affinity-purified His6Tim54 as an antigen. In brief, the predicted mature Tim54 open reading frame starting at Lys-15 was cloned downstream of the His6 tag encoded in the pET28a plasmid (Novogene) and transformed in BL21(RIL) *Escherichia coli*. A 1 L culture was induced with 0.5 mM IPTG at 30 °C for 4 h, and the collected bacterial pellet (3020 × *g* for 10 min) was washed with 0.9% (w/v) NaCl and stored at −20 °C until purification was performed. The bacterial pellet was resuspended in 40 ml lysis buffer (50 mM NaH$_2$PO$_4$, 300 mM NaCl, 10 mM Imidazole, 0.1 mM EDTA, pH 8.0), lysozyme added for 1 mg/ml, and incubated with rocking for 30 min at 4 °C. The bacterial suspension was ruptured using an Avestin Homogenizer, and the resulting lysate was centrifuged at 10,000 × *g* at 4 °C for 20 min. The pellet was solubilized with freshly made Inclusion Body solubilization buffer (1.67%(w/v) Sarkosyl, 10 mM DTT, 10 mM Tris-Cl pH 7.4, 0.05% (w/v) PEG3350) by vortexing on high and then incubated on ice for 20 min. 10 ml of 10 mM Tris-Cl pH 7.4 was added to the suspension, which was then centrifuged at 12,000 × *g* at 4 °C for 10 min. The recovered material was subjected to ammonium sulfate precipitation, with the recombinant Tim54 crashing out of solution at 20% (w/v) ammonium sulfate, and the resulting pellet was resuspended in 2 ml of Inclusion Body solubilization buffer base (10 mM DTT, 10 mM Tris-Cl pH 7.4, 0.05% (w/v) PEG3350) containing 0.5%(w/v) Sarkosyl. To this suspension, 8 ml of Wash Buffer A (0.1% (w/v) Sarkosyl, 50 mM NaH$_2$PO$_4$, 300 mM NaCl, 20 mM Imidazole, 10% (v/v) glycerol, pH 8.0) was added and the solution nutated for 30 min at room temperature. His6Tim54 was then purified from this suspension by incubating with 3 ml Ni-NTA in a capped column rotating for 2 h at room temperature. Following two column-volume washes with 1) 0.1% (w/v) Sarkosyl, 50 mM NaH$_2$PO$_4$, 300 mM NaCl, 20 mM Imidazole, 10% (v/v) glycerol, pH 8.0; and 2) 0.1% (w/v) Sarkosyl, 50 mM NaH$_2$PO$_4$, 600 mM NaCl, 20 mM Imidazole, 10% (v/v) glycerol, pH 8.0, bound material was recovered with elution buffer (250 mM imidazole, 0.1% (w/v) Sarkosyl, 50 mM NaH$_2$PO$_4$, 300 mM NaCl, and 10% glycerol, pH 8.0; 6 sequential 0.5 ml elutions). Protein-containing fractions, identified using the Bradford Assay (Bio-Rad), were combined, PBS dialyzed, and quantified using a BSA standard curve prior to antibody generation.

## Mitochondrial isolation

Isolation of mitochondria from yeast cells was performed as previously described (Calzada et al, 2019). For mitochondrial isolation from T-REx-293 cells, cells were seeded onto three or more 150 mm × 25 mm tissue-culture dishes and allowed to expand in a glucose-based medium supplemented with 0.25 µg/ml doxycycline. Forty-eight hours before isolation, the cells that reached 70–80% confluency were fed fresh medium containing 0.25 µg/ml doxycycline. Mitochondrial isolation was performed according to a previous protocol (Frezza et al, 2007). Briefly, cells were homogenized with IBc buffer [200 mM sucrose, 10 mM Tris-MOPS,

1 mM EGTA/Tris, pH 7.4)] using a Teflon Potter-Elvehjem motor-driven dounce set at 1600 rpm. Homogenates were centrifuged twice at 600 × *g* for 10 min at 4 °C to precipitate debris and the nuclear fraction. The supernatant was centrifuged at 7000 × *g* for 10 min at 4 °C. The resulting pellet was resuspended in IBc buffer and centrifuged at 7000 × *g* for 10 min at 4 °C. This was repeated at 10,000 × *g* for 10 min at 4 °C. The final pellet was resuspended in IBc buffer. If not used immediately, pellets were aliquoted, snap-frozen with liquid N$_2$, and stored at −80 °C for downstream analyses.

## Aac2 membrane topology

About 50 µg of intact mitochondria (in 1 ml BB7.4 [0.6 M sorbitol, 20 mM K + HEPES (pH 7.4)]) or mitoplast were incubated with or without proteinase K (40 µg/ml) on ice. Mitoplasts were obtained by osmotic swelling; to mitochondria suspended in 0.05 ml BB7.4, 19X volumes of 20 mM K + HEPES, pH 7.4 (total 1 ml) was added. After 30 min incubation on ice, 5 mM PMSF was added to deactivate proteinase K. The samples were precipitated by TCA, heated at 60 °C for 5 min, and kept on ice for 5 min for recovery. Following centrifugation at 21,000 × *g* for 20 min at 4 °C, the pellet was washed with cold acetone and resuspended in 0.1 M NaOH. An equal volume of 2X reducing buffer was added and then boiled for 5 min at 95 °C. The samples were analyzed by SDS-PAGE followed by immunoblotting.

## Purification of Aac2 for native MS analysis

8 mg mitochondrial membranes were thawed on ice and reconstituted in BB7.4 (0.6 M sorbitol in 20 mM HEPES buffer) containing 40 µM CATR (Sigma C4992) for 20 min at 4 °C. After centrifuging at 21,000 × *g*, the membrane pellets were solubilized in 1 ml lysis buffer (20 mM Tris pH 8.0, 150 mM NaCl, 10% glycerol, protease inhibitor cocktail, 2% UDM (Anatrace U300LA) on a rotary shaker for 30 min. Solubilized mitochondrial proteins were obtained by centrifuging at 21,000 × *g* for 30 min at 4 °C. Anti-Flag magnetic beads were equilibrated with lysis buffer by washing twice and incubated with the solubilized membrane protein for 2 h on a rotary shaker at 4 °C. The lysate was then aspirated and washed once with lysis buffer and twice with wash buffer (20 mM Tris, pH 8.0, 150 mM NaCl, 10% glycerol, protease inhibitor cocktail, 0.06% UDM). Flag-tagged Aac2 was eluted by incubating with 1 ml elution buffer (20 mM Tris pH 8.0, 150 mM NaCl, 10% glycerol, 0.6% UDM, 125 µg Flag peptide) for 1 h in a rotary shaker at 4 °C. FlagAac2 was concentrated to 5 µM and stored in −80 °C until further use for MS analysis. Purifications were done from two different preparations, with three biological replicates.

## Native MS

Prior to MS analysis, UDM solubilized FlagAac2 were buffer-exchanged into 200 mM ammonium acetate at pH 8.0, with 2 × CMC (critical micelle concentration) of the LDAO (Anatrace D360) detergent using a 7 kDa Zeba spin desalting columns (Thermo). The LDAO exchanged FlagAac2 was then introduced directly into the mass spectrometer using gold-coated capillary needles (prepared in-house). Data were collected on a UHMR mass spectrometer (Thermo Fisher Scientific) optimized for analysis of high-mass complexes, using

methods previously described (Gault et al, 2016). The instrument parameters used were as follows: Positive polarity, capillary voltage 1.2 kV, quadrupole selection from 1000 to 15,000 m/z range, S-lens RF 100%, argon UHV pressure $1.12 \times 10^{-9}$ mbar, temperature 200 °C, resolution of the instrument 17,500 at m/z = 200 (a transient time of 64 ms) and ion transfer optics (injection flatapole, inter-flatapole lens, bent flatapole, transfer multipole: 8, 7, 6, and 4 V, respectively). The noise level was set at 3 rather than the default value of 4.64. In-source trapping of −150 eV was used to release the protein out of the detergent micelles. No collisional activation in the HCD cell was applied at any stage. All other data were visualized and exported using Xcalibur 4.1.31.9 (Thermo Scientific). The relative intensities of apo, CATR, and CL-bound species were obtained by deconvoluting the native MS data using UniDec (Marty et al, 2015). The intensities were converted to mole fractions to determine the fractional populations of each ion in the spectra. Similar parameters were used for data processing in UniDec when comparisons are made. For MSMS experiments, the highest intensity charge state was selected in the quadrupole and subjected to HCD from 50–300 V. Each biological replicate was analyzed with at least two technical replicates.

PE (palmitoyl-oleoyl; Anatrace P416), PG (palmitoyl-oleoyl; Anatrace P616), lyso-PC (palmitoyl; Avanti 855675), and CL (tetramyristoyl; Avanti 710332) were used for lipid add-back experiments. Phospholipids were prepared according to the method described before (Laganowsky et al, 2013). In brief, stock solutions of 3 mM phospholipids were prepared in 200 mM ammonium acetate and stored at −20 °C until use. These were further diluted into 200 mM ammonium acetate containing $2 \times$ CMC of LDAO before titrations and used for further experiments. For titrations, 2.5 μM FlagAac2 protein eluted from $crd1\Delta$ strain were incubated with the required amount of lipid at different concentration points and immediately sprayed into the mass spectrometer.

## Blue native-PAGE

One-dimensional (1D) blue native-PAGE was performed as previously described (Claypool et al, 2006). Briefly, mitochondria were solubilized in lysis buffer [20 mM Tris-Cl, 10% (v/v) glycerol, 100 mM NaCl, 20 mM imidazole, 1 mM CaCl$_2$ (pH 7.4)] supplemented with protease inhibitors (1 mM PMSF, 2 μM pepstatin A, 10 μM leupeptin) containing either 1.5% (w/v) digitonin (Biosynth D-3200) or 2% (w/v) UDM. For pretreatment of yeast mitochondria with CATR or BKA, mitochondria were incubated in BB7.4 with 40 μM CATR (Sigma C4992) or BB6.0 [0.6 M sorbitol, 20 mM KOH-MES (pH6.0)] with 10 μM BKA (Sigma B6179) for 15 min on ice as reported previously (Senoo et al, 2020), pelleted at $21,000 \times g$ for 5 min and then solubilized as above. For human cell mitochondria, IBc pH 7.4 for CATR or pH6.0 for BKA was used during incubation. Protein extracts were collected as supernatant following centrifugation ($21,000 \times g$ for 30 min, 4 °C) and mixed with 10X blue native-PAGE sample buffer [5% (w/v) coomassie brilliant blue G-250 (Serva), 0.5 M 6-aminocaproic acid, and 10 mM bis-tris/HCl (pH 7.0)]. The extracts were resolved by 6 to 16% or 5 to 12% house-made 1D blue native-PAGE gels.

## ADP/ATP exchange

ATP efflux from yeast mitochondria was measured according to previous reports (De Marcos Lousa et al, 2002; Passarella et al, 1988;

Hamazaki et al, 2011) with a slight modification. Isolated mitochondria were suspended in reaction buffer [0.6 M mannitol, 0.1 mM EGTA, 2 mM MgCl$_2$, 10 mM KPi, 10 mM Tris-HCl (pH 7.4)] with ATP detection system containing 5 mM α-ketoglutarate (Sigma-Aldrich 75892), 0.01 mM Ap5A (Sigma D6392), 2 mM glucose, hexokinase (2U/reaction, Sigma H4502), glucose-6-phosphate dehydrogenase (2U/reaction, Sigma-Aldrich G5885), and 0.2 mM NADP (Roche 10128031001). When indicated, 5 mM malate (Sigma M1000) and 5 mM pyruvate (Sigma S8636) were added to the reaction buffer. The exchange reaction was performed on a 96-well plate. Following the addition of external ADP (Sigma A2754), NADPH formation was monitored as increasing absorbance at 340 nm using a plate reader. 10 or 20 μg of mitochondria per well were loaded in the presence or absence of malate and pyruvate, respectively. The initial linear part of the reaction curve was used to calculate the ATP efflux velocity. The kinetics were analyzed by the Michaelis–Menten equation using Graphpad Prism.

## Oxygen consumption measurement in yeast mitochondria

OCR was measured with isolated yeast mitochondria using a Seahorse FluxAnalyzer. Mitochondria were reconstituted in assay buffer [0.25 M sucrose, 5 mM KH$_2$PO$_4$, 2 mM HEPES, 2.5 mM MgCl$_2$, 2 mM EGTA (pH 7.2)] containing fatty acid-free BSA (0.2%). About 50 μl of the reconstituted mitochondria were loaded onto an XF96e assay plate (1.8 μg per well for NADH assay, 3 μg per well for succinate assay) and the assay plate was centrifuged at $2000 \times g$ for 10 min at 15 °C to enhance adherence. Then, an additional 120 μl of assay buffer per well was carefully layered. The compounds were reconstituted in assay buffer without BSA and loaded into the cartridge ports with the following positions at the indicated final concentrations: port A, 5 mM NADH (Millipore 481913) or 10 mM succinate (Alfa Aesar 41983); port B, 1 mM ADP; port C, 10 μM oligomycin (APExBio C3007); port D, 10 μM CCCP (Sigma-Aldrich C2759). OCR was monitored before the first injection and upon each injection with 3 cycles of 1 min-mixing followed by 3 min-measurement at 37 °C.

## Oxygen consumption measurement in human cells

Cells were seeded onto an XF96e cell culture microplate coated with 0.001% (w/v) poly-L-lysin (Sigma-Aldrich P4707) at 30,000 cells/well and incubated for 48 h in glucose-based media supplemented with 0.25 μg/ml doxycycline. On the day of measurement, cells were washed twice with Seahorse XF DMEM Basal Medium supplemented with 10 mM glucose, 2 mM L-glutamine, and 1 mM sodium pyruvate and preincubated for 1 h in a 37 °C humidified CO$_2$-free incubator. OCR was measured by a Seahorse XF96e FluxAnalyzer with the Mito Stress Test kit (Agilent). The Mito Stress Test inhibitors were injected during the measurements as follows; oligomycin (2 μM), FCCP (0.5 μM), rotenone, and antimycin A (0.5 μM). The OCR values were normalized to cell density determined by the CyQUANT Cell Proliferation Assay Kit (Invitrogen C7026) according to the manufacturer's instructions.

## Immunoprecipitation

As indicated above, mitochondria (250 μg) were pretreated with 40 μM CATR in BB7.4. As performed previously (Senoo et al,

2020), the treated mitochondria were pelleted at 21,000 × *g* for 5 min at 4 °C and solubilized with lysis buffer [20 mM Tris-Cl (pH 7.4), 10% (v/v) glycerol, 100 mM NaCl, 20 mM imidazole, 1 mM CaCl₂] supplemented with protease inhibitors (1 mM PMSF, 2 µM pepstatin A, 10 µM leupeptin) containing 1.5% (w/v) digitonin for 30 min at 4 °C. The extracts were clarified by centrifugation at 21,000 × *g* for 30 min at 4 °C, transferred into tubes containing FLAG resin (GenScript), and rotated for 2 h at 4 °C. Post-binding, the resin was sequentially washed with wash buffer [0.1% (w/v) digitonin, 20 mM Tris-Cl (pH 7.4), 100 mM NaCl, 20 mM imidazole, 1 mM CaCl₂], high-salt wash buffer [0.1% (w/v) digitonin, 20 mM Tris-Cl (pH 7.4), 250 mM NaCl, 20 mM imidazole, 1 mM CaCl₂], and once again with wash buffer. Resin-bound proteins were released by boiling in 1X reducing sample buffer and loaded onto 10 to 16% SDS-PAGE gels.

For the SDS-PAGE gel analysis, 1 mg of mitochondria were lysed, followed by immunoprecipitation using FLAG resin as described above. The proteins bound to the resin were eluted by incubating with FLAG peptide (Sigma F3290) in the wash buffer for 30 min at 4 °C. The eluate was precipitated using TCA, resuspended in 0.1 M NaOH, and mixed with an equal volume of 2X reducing buffer. The samples were then boiled for 5 min and loaded onto a 10–16% SDS-PAGE gel. The gel was stained with SYPRO Ruby (Invitrogen S12000) according to the manufacturer's protocol.

## Complex III and IV activity measurements

Respiratory complex III and IV activities were measured as previously described (Tzagoloff et al, 1975; Dienhart and Stuart, 2008). Briefly, mitochondria were solubilized in 0.5% (w/v) *n*-dodecyl-β-D-maltoside (DDM, Sigma-Aldrich) spiked with protease inhibitors (1 mM PMSF, 2 µM pepstatin A, 10 µM leupeptin). To initiate the Complex III reaction, 50 µg of mitochondria and 100 µM decylubiquinol were added to reaction buffer (50 mM KPi, 2 mM EDTA, pH 7.4) containing 0.008% (w/v) horse heart cytochrome c (Sigma) and 1 mM KCN. Complex IV reaction was initiated by adding 15 µg mitochondria to the reaction buffer with 0.008% (w/v) ferrocytochrome c. The reduction (for complex III) or oxidation (for complex IV) was followed at 550 nm using a spectrophotometer.

## Complex V in gel activity

Complex V in gel activity was performed in accordance with previous reports (Zerbetto et al, 1997; Wittig et al, 2007). 100 µg mitochondria were solubilized in 1% (w/v) DDM (Sigma-Aldrich) in lysis buffer [20 mM Tris-Cl (pH 7.4), 10% (v/v) glycerol, 100 mM NaCl, 20 mM imidazole, 1 mM CaCl₂] spiked with protease inhibitors (1 mM PMSF, 2 µM pepstatin A, 10 µM leupeptin). The lysates were resolved by 5–12% blue native-PAGE gels. Complex V in gel activity was performed by incubating the gels in assay solution [270 mM glycine, 35 mM Tris-Cl, pH 8.3, 14 mM MgSO₄, 0.2% (w/v) Pb (NO₃)₂, 8 mM ATP] and stopped by replacing the solution with 50% methanol. Subsequently, the gel was stained with Coomassie.

## Statistical analysis

Immunoblots were quantified using Image J and ImageLab (BIO-RAD) software. Statistical analyses were performed using Graphpad Prism (ver. 9.3). The statistical tests performed, sample sizes, and determined *P* values are provided in the figure or its accompanying legend. Representative blot images from at least three independent experiments performed on at least three separate days are presented in the figures.

## Simulation methods

### System generation

MD simulations and FEP calculations were employed to quantify the binding affinity between the Human ANT1 protein and CL. The crystal structure of the Human ANT1 protein has not been experimentally determined, therefore, a full atomic model of the Human ANT1 protein was constructed using the homology module (Prime) of the Schrödinger suite(Schrödinger Release 2023-1: Prime, Schrödinger, LLC, New York, NY, 2021.; Jacobson et al, 2002, 2004). The sequence alignment of Human ANT1 with Bovine ANT1 and Yeast AAC2 is shown in Appendix Fig. S16A. Human ANT1 is highly homologous to Bovine ANT1 (96% sequence identity), while it shares 50–54% sequence identify with the Yeast AAC proteins (AAC1-3). Consequently, the crystal structure of ANT1 from Bos taurus (PDB ID: 1OKC) was used as a homology modeling template, and the Schrödinger prime was used to build the Human model. Structural Analysis and Validation server (SAVES - https://saves.mbi.ucla.edu/) and PROCHECK (Laskowski et al, 1993) was used to validate the generated homology model. Many steps of loop refinement using Schrödinger (Schrödinger Release 2023-1: Prime, Schrödinger, LLC, New York, NY, 2021.) and solvent minimization (GROMACS 2020.4 (Abraham et al, 2015)) were performed to improve the overall quality of the structure. The final obtained high resolution (quality factor 97%) Human ANT1 protein structure is shown in Appendix Fig. S16. The quality of the obtained 3D model was assessed using the Ramachandran plot (PROCHECK) (Appendix Fig. S16D). It is observed that none of the residues were lying in the disallowed regions, indicating the predicted Human ANT1 model is high quality. The Human ANT1 protein was inserted into a bilayer composed of 80% POPC (1-palmitoyl-2-oleoyl-*sn*-glycero-3-phos-phocholine) and 20% CL (TLCL⁻²) lipids (1'-[1,2-dilinoleoyl-*sn*-glycero-3-phospho]-3'-[1,2-dilinoleoyl-*sn*-glycero-3-phospho]-gly-cerol) using the CHARMM-GUI membrane builder module(Jo et al, 2008; Wu et al, 2014). TIP3P water molecules were used to solvate the system and potassium ions were added to neutralize the total charge of the system. A sample representation of the system setup is shown in Fig. S11.

### Molecular dynamics (MD) simulation protocol

All-atom MD simulations were performed using GROMACS 2020.4 (Abraham et al, 2015) with CHARMM36 force field parameters for protein and lipid molecules (Huang et al, 2017; Lee et al, 2016). CHARMM-GUI generated protocol was implemented to equilibrate the system. In detail, 20,000 steps of steepest descent energy minimization were performed, followed by two short NVT simulations of 0.5 ns each, with a time step of 1 fs. Harmonic positional restraints were employed on the protein and lipid atoms during the NVT simulations; the force constants were reduced in the second NVT simulation, Subsequently, four steps of restrained NPT equilibrium simulations (0.5 ns each) were performed, where the restraints were reduced/released in each

successive simulation step. The time step was 1 fs in the first NPT simulation and 2 fs in all subsequent simulations. The Berendsen thermostat (Berendsen et al, 1984) and barostat (Feenstra et al, 1999) were used to maintain the system at 300 K and 1 bar pressure with a coupling constant of 1 and 5 ps, respectively. Production NPT simulations were performed for 1 μs with a time step of 2 fs. The coordinates of the trajectory were saved for every 50 ps. Nosé-Hoover thermostat (Nosé, 1984) and a Parrinello–Rahman barostat (Parrinello and Rahman, 1981) were used to maintain the temperature and pressure of the system at 300 K and 1 bar, respectively. The temperature and pressure coupling time constants were set to 1 and 5 ps, correspondingly. Semi-isotropic pressure coupling was employed with the compressibility factor of $4.5 \times 10 - 5 \, bar^{-1}$. Long-range electrostatics were calculated with the Particle-Mesh Ewald (PME) method(Darden et al, 1993; Essmann et al, 1995) and the short-range van der Waals and electrostatic interactions were calculated with a cut-off distance of 12 Å and used a switching distance of 10 Å. The LINCS algorithm (Hess et al, 1997) was used to constrain the hydrogen atoms involved in covalent bonds.

To understand the effects of ANT1 protein mutations on the binding of CL, two human ANT1 mutant systems were studied in which the lysine 141 residue was mutated to phenylalanine (L141F) or glutamic acid (L141E). The final protein configuration from the 1-μs wild-type (WT) Human ANT1 simulation was used to generate the L141F and L141E systems using CHARMM-GUI. The mutant proteins were then inserted into a POPC:TLCL$^{-2}$ (80:20) bilayer using CHARMM-GUI, and the same MD protocol was used to equilibrate and simulate the mutant systems as the WT protein. All the performed simulations are detailed in Table S4. The figures were generated using VMD (Humphrey et al, 1996) and PyMOL (The PyMOL Molecular Graphics System, Version 2.0 Schrödinger, LLC). It was noticed that some CL lipid molecules occupied the known binding sites of AAC/ANT1 proteins (Hedger et al, 2016; Corey et al, 2019; Mao et al, 2021; Duncan et al, 2018) in the equilibrated systems (prior to production MD). Therefore, we also ran a set of equilibrium simulations in which the prebound CL lipid molecules were removed from pocket 2 and simulated for 1 μs using the above-detailed MD protocol for both the WT and mutant proteins. Therefore, we describe a set of simulations in which CL is initially bound at pocket 2 (termed "prebound") and a set of simulations in which the CL initially bound at pocket 2 has been removed (termed "unbound").

### Free energy perturbation (FEP) calculations

A representative frame (t ≈ 0.965 μs) with CL$^{-2}$ within the pocket 2 binding site was selected from the WT prebound simulation as the input for the FEP calculations. Similarly, the corresponding equilibrium protein structure (1 μs) was used to generate the mutant FEP systems, i.e., L141F and L141E. For these calculations, the bound CL$^{-2}$ lipid in pocket 2 was renamed to LIG (for ligand), and the same ligand conformation was used as a starting structure for all the simulated systems. The target lipid (LIG) was decoupled from the simulation box in the presence ("protein–ligand–membrane complex") and or the absence of the Human ANT1 protein ("ligand–membrane complex"). To maintain charge neutrality during the decoupling of LIG, which carried a −2 charge, a Ca$^{+2}$ ion was also decoupled from the system, so that both

CL$^{-2}$ and Ca$^{+2}$ are being decoupled simultaneously. The free energies were estimated using the alchemical thermodynamic cycle described in Fig. S13. All FEP simulations were run in GROMACS 2020.4 (Abraham et al, 2015), and a linear alchemical pathway was employed to decouple (λ = 0 to λ = 1) the LIG Lennard-Jones and coulombic interactions sequentially (Aldeghi et al, 2016). The Lennard-Jones interactions were transformed with Δλ = 0.05 and the charges were annihilated through coulombic transformations with Δλ = 0.1. The LIG molecule was restrained, to maintain its relative position and orientation with respect to the ANT1 protein (Corey et al, 2019; Aldeghi et al, 2016) through two distances, two angles, and three dihedral harmonic potentials. Similar to Aldeghi et al calculations, the above mentioned restraints were transformed using 12 non-uniformly distributed λ values for the protein–ligand–membrane complex systems (Aldeghi et al, 2016). For the ligand–membrane system, the free energy contribution of the restraints to the binding free energy was estimated analytically to be 8.311 kcal/mol. (Corey et al, 2019; Aldeghi et al, 2016; Boresch et al, 2003) A total of 42 windows for the protein–ligand–membrane complex simulations and 31 windows for the ligand–membrane complex simulations were performed. For each window, 20,000 steps of steepest descent energy minimization were performed. Subsequently, all the windows were simulated for 0.5 ns under NVT ensemble conditions. The positional restraints were employed with a force constant of 500 and 100 kJ mol$^{-1}$ nm$^{-2}$ on the protein and lipid heavy atoms, correspondingly. Following NVT, a 1 ns position restrained NPT simulation was performed using Berendsen barostat (Feenstra et al, 1999). A position restraint force of 50 kJ mol$^{-1}$ nm$^{-2}$ was applied on the protein and lipid heavy atoms and the pressure was set to 1 bar. Further, 0.5 ns of unrestrained NPT simulation was performed and the Parrinello–Rahman barostat (Parrinello and Rahman, 1981) was used to maintain the total pressure of the system. Finally, each window was simulated under NPT ensemble conditions for a production run of 15 ns. A semi-isotropic pressure coupling was performed with a coupling constant of 5 ps using the Parrinello–Rahman barostat. Langevin dynamics integrator was employed with a time-step of 2 fs. The PME algorithm was used with a Fourier grid spacing of 1.2 Å to calculate the long-range electrostatic interactions (Darden et al, 1993; Essmann et al, 1995). The short-range non-bonded interactions were calculated with a cut-off distance of 12 Å and a switching distance of 10 Å. A PME spline order of 6 was employed with a relative tolerance set to 10$^{-6}$. The soft-core potential (sigma = 0.3) was used to transform the Lennard-Jones interactions (Corey et al, 2019; Aldeghi et al, 2016). The GROMACS implemented long-range dispersion correction for energy and pressure was applied (Aldeghi et al, 2016). Lincs algorithm was used to constrain the H-bonds (Hess et al, 1997). The Alchemical Analysis package (Klimovich et al, 2015; Shirts and Chodera, 2008) was used to estimate the free energy from the individual windows. The final 10 ns of simulation data was used to estimate the free energy values. Multistate Bennet acceptance ratio (MBAR) was used in the pymbar package to calculate free energies (Shirts and Chodera, 2008). The entire FEP calculations were repeated four times for each simulated system, i.e., Human ANT1 WT, L141F, and L141E systems. The final binding free energies were estimated using Eq. 1. Data from three repeats were averaged, and the standard errors were reported (Aldeghi et al, 2016).

Wilcoxon rank-sum tests were performed to assess the statistical significance of the binding free energies between WT and mutants.

$$\Delta G_b = -\Delta G_{\text{elec}+vdW+\text{restr}}^{\text{complex}} + \Delta G_{\text{elec}+vdW}^{\text{solv}} + \Delta G_{\text{rest\_on}}^{\text{solv}} \qquad (1)$$

## Data availability

This study includes no data deposited in external repositories.

The source data of this paper are collected in the following database record: biostudies:S-SCDT-10_1038-S44318-024-00132-2.

## Peer review information

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

## Acknowledgements

We thank Carla Koehler (UCLA) and Jean-Paul Lasserre (University of Bordeaux) for the generous gifts of antibodies. This work was supported in part by NIH grants (R01HL108882 and R01HL165729 to SMC, R35GM119762 to ERM, Biochemistry, Cellular, and Molecular Biology Program Training Grant T32GM007445 for MGB and MSTP Training Grant T32GM136577 for JAS), pre-doctoral fellowships from the American Heart Association (10PRE3280013 to MGB and 15PRE24480066 to OBO), a post-doctoral fellowship from the Uehara Memorial Foundation (to NS), a post-doctoral fellowship from the

American Heart Association and the Barth Syndrome Foundation (Award ID: 828058 to NS), and Royal Society Newton international fellowship (NIF \R1\181108 to BS and NIF\R1\192285 to DKC).

## Author contributions

**Nanami Senoo**: Conceptualization; Formal analysis; Funding acquisition; Investigation; Visualization; Methodology; Writing—original draft; Writing—review and editing. **Dinesh K Chinthapalli**: Formal analysis; Funding acquisition; Investigation; Methodology; Writing—original draft; Writing—review and editing. **Matthew G Baile**: Conceptualization; Funding acquisition; Investigation; Writing—review and editing. **Vinaya K Golla**: Formal analysis; Investigation; Methodology; Writing—original draft; Writing—review and editing. **Bodhisattwa Saha**: Formal analysis; Funding acquisition; Investigation; Methodology; Writing—original draft; Writing—review and editing. **Abraham O Oluwole**: Formal analysis; Investigation; Writing—review and editing. **Oluwaseun Ogunbona**: Funding acquisition; Investigation; Writing—review and editing. **James A Saba**: Investigation; Methodology; Writing—review and editing. **Teona Munteanu**: Investigation; Methodology. **Yllka Valdez**: Investigation; Writing—review and editing. **Kevin Whited**: Investigation. **Macie S Sheridan**: Investigation. **Dror Chorev**: Methodology; Project administration; Writing—review and editing. **Nathan N Alder**: Supervision; Funding acquisition; Project administration; Writing—review and editing. **Eric R May**: Formal analysis; Supervision; Funding acquisition; Methodology; Writing—original draft; Project administration; Writing—review and editing. **Carol V Robinson**: Conceptualization; Formal analysis; Supervision; Funding acquisition; Investigation; Methodology; Writing—original draft; Project administration; Writing—review and editing. **Steven M Claypool**: Conceptualization; Formal analysis; Supervision; Funding acquisition; Investigation; Methodology; Writing—original draft; Project administration; Writing—review and editing.

Source data underlying figure panels in this paper may have individual authorship assigned. Where available, figure panel/source data authorship is listed in the following database record: biostudies:S-SCDT-10_1038-S44318-024-00132-2.

## Disclosure and competing interests statement

The authors declare no competing interests.

# Expanded View Figures

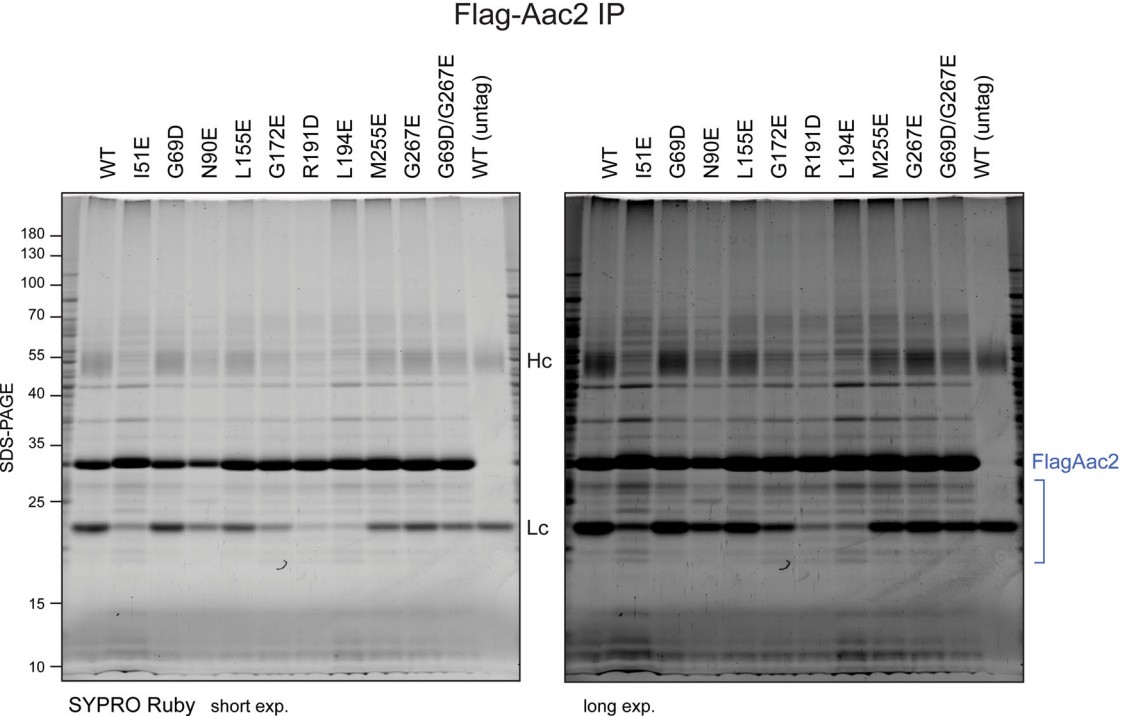

**Figure EV1.  Aac2 CL-binding mutants do not engage in aberrant protein interactions.**

Isolated mitochondria were solubilized with 1.5% digitonin and subjected to FLAG immunoprecipitation. Co-purified extracts were resolved by 10–16% SDS-PAGE and resolved proteins detected by SYPRO Ruby staining.

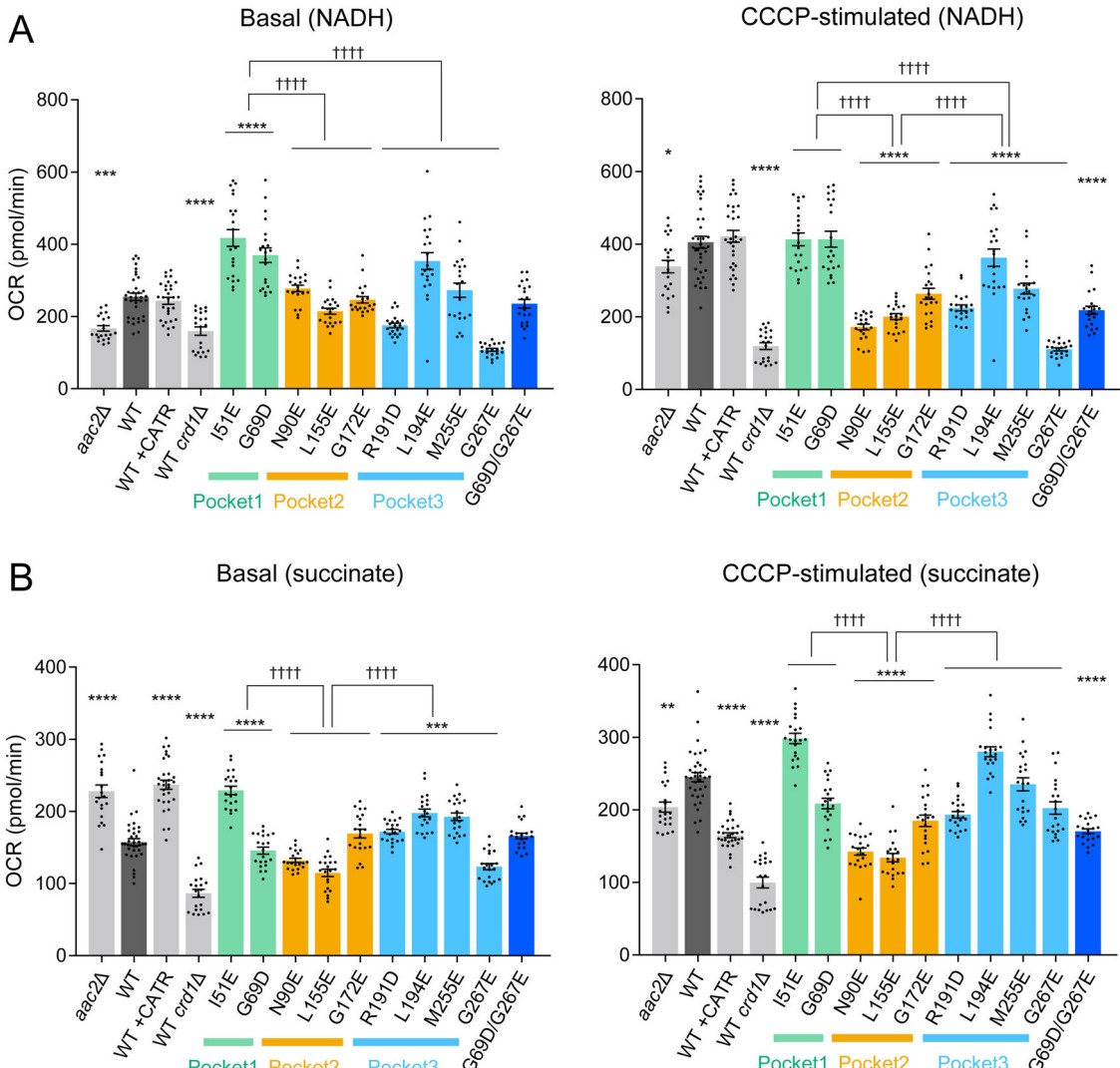

**Figure EV2. Mitochondrial respiration of Aac2 CL-binding mutants.**

Related to Fig. 4D–F, basal and CCCP-stimulated respirations of WT and mutant mitochondria in the presence of NADH (**A**) and succinate (**B**) were plotted as oxygen consumption rate (OCR) ($n = 21$–35, 3–5 biological replicates with 5–7 technical replicates). Mean with SEM. Significant differences obtained by two-way ANOVA followed by Tukey's multiple comparisons test are shown as * for comparison with WT and † for comparison between pockets; *$p < 0.05$, **$p < 0.01$, ***$p < 0.001$, ****$p < 0.0001$.

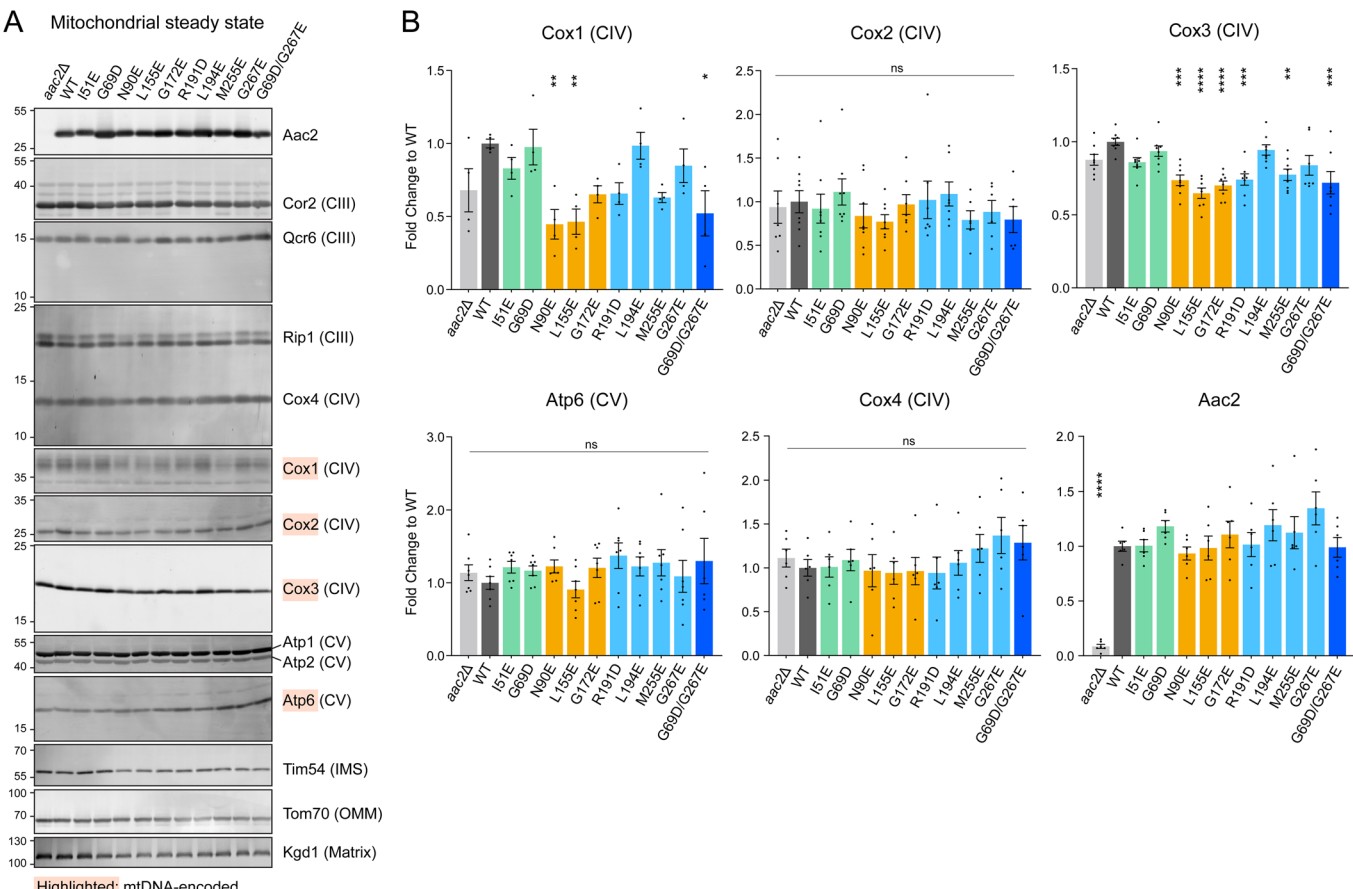

**Figure EV3.  The expression of respiratory complex subunits encoded in mitochondrial DNA is attenuated in Aac2 CL-binding mutants.**

(A) Mitochondrial extracts were resolved by SDS-PAGE and immunoblotted for indicated proteins, including subunits of respiratory complexes III, IV, and V. (B) The expression of indicated respiratory complex subunits was quantified. Mean with SEM. Statistical differences were analyzed by one-way ANOVA followed by Dunnett's multiple comparison test; *$p < 0.05$, **$p < 0.01$, ***$p < 0.001$, ****$p < 0.0001$ (vs. WT). Representative images from the replicates ($n = 4$–8, biological replicates) are shown.

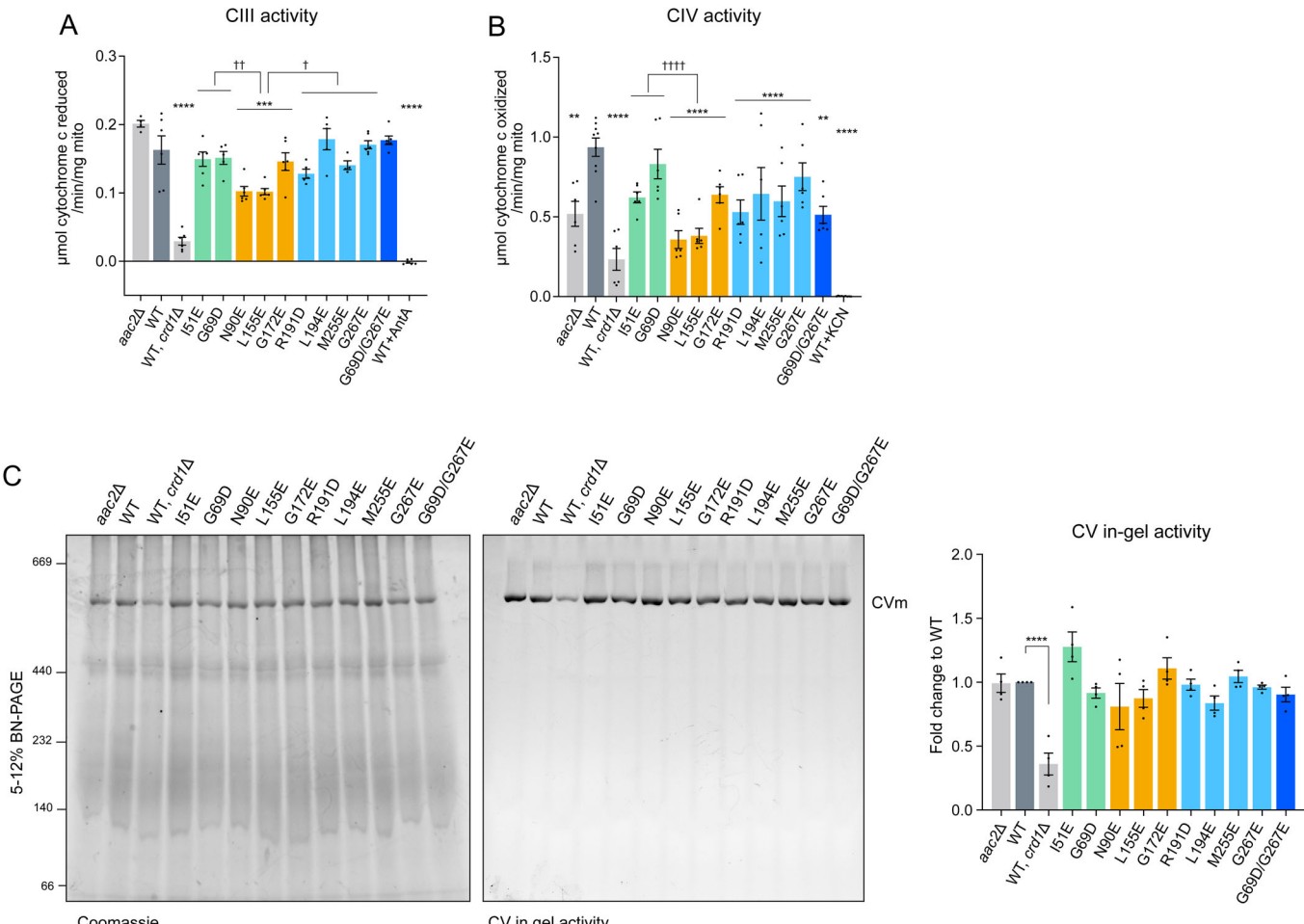

**Figure EV4. Activities of respiratory complexes III, IV, and V of CL-binding mutants.**

(**A**) Complex III activity in 0.5% (w/v) DDM-solubilized mitochondria ($n = 4$–6, biological replicates). (**B**) Complex IV activity in 0.5% (w/v) DDM-solubilized mitochondria ($n = 6$–9, biological replicates). (**C**) Complex V in-gel activity assay. Mitochondria were solubilized in 1% (w/v) DDM, resolved by 5–12% blue native-PAGE, and incubated with the substrate ($n = 5$, biological replicates). Mean with SEM. Significant differences obtained by two-way ANOVA followed by Tukey's multiple comparisons test are shown as * for comparison with WT and † for comparison between pockets; *$p < 0.05$, **$p < 0.01$, ***$p < 0.001$, ****$p < 0.0001$.

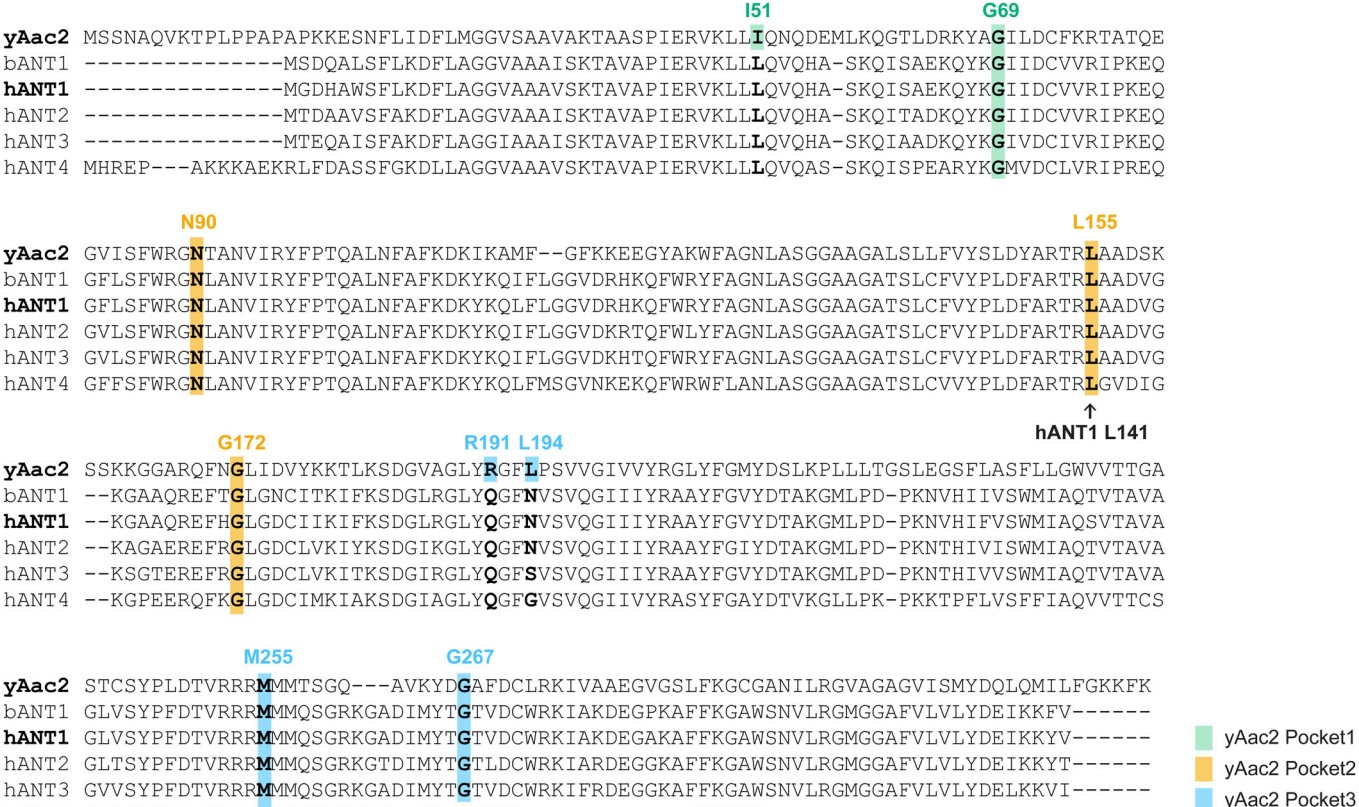

**Figure EV5. CL-binding sites are conserved across species.**

Amino acid sequence alignment of yeast Aac2, bovine ANT1, and human ANT isoforms. The residues designed for the Aac2 CL-binding mutants are highlighted as indicated.

