## [Peer Review File · The EMBO Journal]

Functional diversity among cardiolipin binding sites on the mitochondrial ADP/ATP carrier

Nanami Senoo, Dinesh Chinthapalli, Matthew Baile, Vinaya Golla, Bodhisattwa Saha, Abraham Oluwole, Oluwaseun Ogunbona, James Saba, Teona Munteanu, Yllka Valdez, Kevin Whited, Macie Sheridan, Dror Chorev, Nathan Alder, Eric May, Carol Robinson, and Steven Claypool

Corresponding author(s): Steven Claypool (sclaypo1@jhmi.edu)

Review Timeline:

Submission Date:	11th Jul 23
Editorial Decision:	29th Aug 23
Appeal Received:	30th Aug 23
Editorial Decision:	30th Sep 23
Revision Received:	21st Mar 24
Editorial Decision:	23rd Apr 24
Revision Received:	3rd May 24
Accepted:	8th May 24

Editor: William Teale

Transaction Report:

Dear Steve,

Thank you again for the submission of your manuscript entitled "Conserved cardiolipin-ADP/ATP carrier interactions assume distinct roles that are clinically relevant" (EMBOJ-2023-114977) to The EMBO Journal. Please accept my sincere apologies for the unusually long peer-review period take for your study. Your manuscript was sent to two reviewers for evaluation, which I enclose below. A third referee withdrew at a late stage, making our decision here in the editorial team a very difficult one. Whilst referee #2 states clearly that your work will make an important contribution to the field, referee #1 raises key concerns that we cannot see being easily addressed by a round of revision. We have decided, therefore that your manuscript is not currently suitable for publication in The EMBO Journal.

You address the functional implications of cardiolipin binding to the ADP/ATP carrier. Using a combination of native-MS and BN-PAGE, you demonstrate that cardiolipin binding stabilises the ADP/ATP carrier's structure and local protein environment; you then go on to demonstrate cardiolipin is necessary for carrier, and overall mitochondrial, function. The effect of the mutations you carry out are, however, wide-ranging (as seen by the destabilization of all three cardiolipin binding sites when only one is mutated). Reviewer 1 concludes that you therefore do not conclusively demonstrate that cardiolipin-pocket mutations are exclusively affecting cardiolipin binding and not other, as yet undescribed, properties of the carrier. This is also borne out by the changes to migration under BN-PAGE in samples prepared from cells in which cardiolipin biosynthesis is genetically inhibited. Therefore, we must conclude that the biological insight you report is, at this stage, not sufficiently well-defined for us to proceed towards publication on this occasion.

I am sorry we cannot be more positive. I nevertheless hope that (whichever direction this work takes towards publication) you will find our referees' comments helpful and continue to see The EMBO Journal as a suitable location to disseminate your work.

Best wishes,

William

William Teale, PhD
Editor
The EMBO Journal
w.teale@embojournal.org

Referee #1:

Senoo et al present an in-depth investigation of the association between the ADP/ATP carrier (AAC) and cardiolipin (CL) in yeast. It has been known for some time that AAC binds 3 CL molecules at specific sites (binding pockets). These sites were previously identified by crystallography. The AAC-CL interaction is deemed functionally important because it stabilizes the tertiary structure of the AAC, an abundant mitochondrial protein that is crucial for oxidative phosphorylation.

In order to establish the mechanism of AAC-CL interactions, the authors created several AAC mutants by substituting amino acids in the 3 binding pockets. They FLAG-tagged the mutated AACs, purified them, and analyzed CL-AAC interactions by native mass spectrometry. This is the most interesting and innovative part of the study. As expected, mutations weakened CL-AAC binding but the effects were not specific for the targeted pockets because most mutations (except I151E & L194E) led to the loss of all 3 CL molecules (Fig 2D). The authors went on to study the physiological effects of the mutations, which turned out to be variable. Some mutations slowed down growth. Most mutations reduced ADP/ATP exchange, oxygen consumption, and respiratory control ratios (RCRs). The mutations also altered AAC migration in BN-PAGE from a distinct band to a smear and reduced the amount of AAC that co-migrated with respiratory supercomplexes. Finally, the authors modeled a myopathy-causing mutation of the human adenine nucleotide carrier in yeast and in 293 cells. They found similar effects of this mutation as described for the "binding pocket mutants". Molecular dynamics simulations suggested that the human mutation altered the structure of binding pocket 2, which reduced the likelihood of CL to reach the binding pocket (on-rate) but did not cause the loss of CL that was already in the pocket (off-rate).

The paper describes an impressive body of work but the key conclusions are not supported by the data. In my mind, it will be difficult to develop a clear-cut mechanistic interpretation because of the intricate relation between CL binding and protein conformation. This leaves us with a mostly descriptive set of data.

MAJOR CRITIQUE

1. The main problem of the manuscript is that it is difficult, if not impossible, to separate local effects on CL binding from distant effects on secondary and tertiary structure. Each mutation probably altered the local environment, which may have caused the loss of the local CL resident, but in addition, each mutation clearly had long-range effects, which destabilized the binding of distant CL molecules and possibly altered other aspects of the protein structure. Therefore, one cannot associate specific functional defects with the loss of distinct CL molecules. Unfortunately, this undermines the main conclusions of the paper, such as "...that the three buried CLs are functionally distinct..." (results) or that "... transport activity was impaired in a binding site-specific manner" (abstract). For any of the mutations, it is not clear whether the loss of CL is the actual cause of protein dysfunction or merely an associated phenomenon. This raises the question of whether or not mutations outside the binding pockets also affect CL binding. Furthermore, different mutations in the same pocket did not always show the same effect, which further discredits the idea of "pocket-specific functions".
2. Interpretation of the BN-PAGE data is problematic. Conventionally, one thinks of the migration behavior of a protein in a BN-PAGE experiment to either reflect its size and charge or that of complexes it binds to. Based on their BN-PAGE data, the authors concluded that AAC mutants are properly folded in the IMM (correctly I believe) but that the mutations changed the "tertiary fold" of the AAC under BN-PAGE conditions. What does that mean? How can a protein be properly folded but at the same time have the wrong tertiary fold? Similarly confusing is the term "monomeric folding structure". Is this supposed to say the protein is a monomer? My interpretation of the BN-PAGE data is that the mutations make the AAC prone to aberrant interactions with other proteins. This may or may not have anything to do with CL.

MINOR ISSUES

3. The abstract does not project a clear message and does not capture the content of the work very well.
4. Fig 2D is very important. Therefore, individual data should be presented in addition to means and standard deviations.
5. The sentence "The respiratory growth capacity of *crd1Δ* and some of the *Aac2* mutants was exacerbated at elevated temperature" is probably trying to say that the growth defect was exacerbated at elevated temperature.
6. In the title, the expression "clinically relevant" is odd because the work is scientifically relevant but has no obvious practical implications for the management of patients with AAC mutations. Perhaps "disease-causing" would be better.

Referee #2:

This is an interesting and well-written manuscript. The flow of the experiments is logical, and the experiments are solid. It will make an important contribution to the field.

- 1) How do different CL species affect the function of the AAC? What is the effect of ALCAT and Delta6 desaturase, which were shown to be changed in metabolic disease, affecting respiratory function? I do not ask for additional experiments but to make this study relevant to disease, it will be valuable to consider the different perturbations to CLs.
- 2) Experimentally, the main shortcoming of the approach is the lack of consideration of ATP synthase. How do the authors differentiate effects on CV (ATP synthase) vs. effects on AAC
- 3) Is the effect of CL on monomerization specific to CL?
- 4) In Figure 3, I still do not understand how the authors can conclude that the effect of the mutations on the tertiary structure is contributed exclusively and entirely by the CL interaction capacity.
- 5) Figure 4. Did the authors consider the possibility that the mutants that had reduced exchange velocities changed confirmation to become an uncoupling channel, as described by Bertholet et al.? Bertholet et al. indeed find a reciprocal relationship between exchange and uncoupling
- 6) In Figure S4, what is the explanation for the high level of variability in this mutant?
- 7) In Figure 4, the authors test the effect of energized as compared to non-energized mitochondria, which is expected to affect membrane potential. However, the uncoupling ability of the AAC is also expected to change membrane potential. Should the study compare membrane potential and uncoupling in the different mutants?
- 8) The statement that ADP-stimulated respiration is a reflection of AAC activity is incorrect. A linear relationship between respiration and ADP availability is limited to a situation where respiration is not limited by the respiratory chain complexes or substrate.
- 9) Fig S5 indicates that similar trends of mutant effects were found with CCCP. However, in the presence of CCCP, the AAC does not play any role in determining respiratory rate. Isn't that an indication that the mutants affect the respiratory chain independent of ADP/ATP transport?
- 10) Authors should at least mention previous studies demonstrating the activity of AAC as a proton transport channel, as shown

by Berthollet et al. If authors consider that H⁺ is also transported by AAC, it will be very interesting to measure or discuss the implications of this activity in their system with their mutants. This could also affect the interpretation of the respiratory experiment, as shown in Figure 6F.

** As a service to authors, EMBO Press provides authors with the possibility to transfer a manuscript that one journal cannot offer to publish to another EMBO publication or the open access journal Life Science Alliance launched in partnership between EMBO Press, Rockefeller University Press and Cold Spring Harbor Laboratory Press. The full manuscript and if applicable, reviewers' reports, are automatically sent to the receiving journal to allow for fast handling and a prompt decision on your manuscript. For more details of this service, and to transfer your manuscript please click on Link Not Available. **

Revision plan for EMBOJ-2023-114977R

Rev 1 Point 1 and Rev 2 Point 4: Are there long-range structural impacts of mutations beyond intended short-range perturbation of CL-binding?

We plan to address this important point in 3 manners:

1. We will analyze 1-microsecond equilibrium MD simulations of WT and mutant (L141F and L141E) ANT1 conducted in a CL containing bilayer. We will compare metrics of global stability (RMSD) and local structural features (secondary structure, local RMSD) between WT and mutants to observe if mutations induce significant structural perturbations.
2. We will analyze cardiolipin binding propensity at non-mutated pocket sites (pocket 1 and pocket 3) in WT and mutant equilibrium simulations. The analyses will encompass lipid binding density and protein-lipid contact probabilities to observe if mutations at pocket 2 (L141F/E) have long range effects on distal CL binding sites.
3. Since the charge state of a protein in a mass spectrum is related to the exposed surface area of a protein it is an exquisitely sensitive proxy for the overall fold of the protein. There are many references to support this statement. It is evident therefore that when there is a particularly large shift in a charge state distribution (CSD) the protein is unfolded. A minor perturbation in folded structure may result in a slight shift in the CSD of one protonation site. With this in mind, we re-analyzed the charge state distributions obtained in native mass spec experiments. We determined that four mutants have slight perturbations to the CSD which we interpret to reflect small changes to their surface area. The remaining six mutants displayed wild type-like charge states. These results will be used to stratify the mutants and contextualize our results.

Rev 1, Point 1: Do mutations outside the binding pockets also affect CL binding?

This is an important point. To test this, we plan to fully characterize 2-3 additional Aac2 mutants that do not target any of the CL-binding pockets. The additional mutants planned include a pathogenic variant, A137D that lacks transport activity but interacts with respiratory complex subunits similar to wild type Aac2 (PMID: 29688796) and two residues, K112D and K215D, that participate in salt-bridge networks in the matrix-open conformational state (these residues are on the cytoplasmic side of the transport pathway; PMIDs: 26453935, 30611538, 24474793).

Rev 1, Point 2: Do the Aac2 mutants engage in aberrant interactions with other proteins?

To test this, we will perform FLAG immunoprecipitations of WT and mutant Aac2s, untreated or pretreated with CATR, and determine if novel protein bands are co-purified for untreated mutant Aac2s. Based on the co-IPs already performed (fig. S8), CATR-treated Aac2 mutants should co-purify complex III and IV subunits in amounts similar to or less than wild type Aac2 (untreated or CATR-bound). Note that this strategy also addresses Rev 1 Point 1 and Rev 2 Point 4.

Rev 2, Point 2: What about the ATP synthase?

We will determine ATP synthase activity in mitochondria expressing wild type Aac2 versus Aac2 mutants. In addition, we will explicitly discuss the limitations of our functional assays which cannot disentangle whether any defects observed are directly tied to a mutant Aac2, as targeted, versus an unexpected impact on the ATP synthase. Long term, we hope to establish an *in vitro* liposome reconstitution work flow to measure the transport activity of Aac2/ANT1 mutants in isolation; however, this is beyond the scope of the current study.

Rev 2, Points 7, 8 and 9: Is the respiratory chain damaged in Aac2 mutants?

We will assess respiratory chain activities. The results will be interpreted in the context of the following results already included in the original submission:

- yAac2 mutants: We already show that the respiratory chain is damaged to some extent, particularly for pocket 2 mutants which had low CCCP respiration and CIV expression.
- hANT1 mutants: Their lower maximal (FCCP) respiration is suggestive of a damaged respiratory chain. To augment this observation, we will assess respiratory chain subunit expression.

Dear Steve,

I have now shared your proposed revision plan with both referees. In each case, they encouraged the work to proceed along the proposed lines. I have therefore re-opened your submission and would like to invite you to submit a revised version of the manuscript.

I would also like to point out that as a matter of policy, competing manuscripts published during this period will not be taken into consideration in our assessment of the novelty presented by your study ("scooping" protection). We have extended this 'scooping protection policy' beyond the usual 3 month revision timeline to cover the period required for a full revision to address the essential experimental issues. Please contact me if you see a paper with related content published elsewhere to discuss the appropriate course of action.

Again, please contact me at any time during revision if you need any help or have further questions.

Thank you very much again for the opportunity to consider your work for publication. I look forward to your revision.

Best regards,

William

William Teale, Ph.D.
Editor
The EMBO Journal

When submitting your revised manuscript, please carefully review the instructions below and include the following items:

2) individual production quality figure files as .eps, .tif, .jpg (one file per figure).

3) a .docx formatted letter INCLUDING the reviewers' reports and your detailed point-by-point response to their comments. As part of the EMBO Press transparent editorial process, the point-by-point response is part of the Review Process File (RPF), which will be published alongside your paper.

4) a complete author checklist, which you can download from our author guidelines ([https://wol-prod-cdn.literatumonline.com/pb-assets/embo-site/Author Checklist%20-%20EMBO%20J-1561436015657.xlsx](https://wol-prod-cdn.literatumonline.com/pb-assets/embo-site/Author%20Checklist%20-%20EMBO%20J-1561436015657.xlsx)). Please insert information in the checklist that is also reflected in the manuscript. The completed author checklist will also be part of the RPF.

6) We require a 'Data Availability' section after the Materials and Methods. Before submitting your revision, primary datasets produced in this study need to be deposited in an appropriate public database, and the accession numbers and database listed under 'Data Availability'. Please remember to provide a reviewer password if the datasets are not yet public (see <https://www.embopress.org/page/journal/14602075/authorguide#datadeposition>). If no data deposition in external databases is needed for this paper, please then state in this section: This study includes no data deposited in external repositories. Note that the Data Availability Section is restricted to new primary data that are part of this study.

Note - All links should resolve to a page where the data can be accessed.

Please remember: Digital image enhancement is acceptable practice, as long as it accurately represents the original data and

conforms to community standards. If a figure has been subjected to significant electronic manipulation, this must be noted in the figure legend or in the 'Materials and Methods' section. The editors reserve the right to request original versions of figures and the original images that were used to assemble the figure.

8) For data quantification: please specify the name of the statistical test used to generate error bars and P values, the number (n) of independent experiments (specify technical or biological replicates) underlying each data point and the test used to calculate p-values in each figure legend. The figure legends should contain a basic description of n, P and the test applied. Graphs must include a description of the bars and the error bars (s.d., s.e.m.).

9) We would also encourage you to include the source data for figure panels that show essential data. Numerical data can be provided as individual .xls or .csv files (including a tab describing the data). For 'blots' or microscopy, uncropped images should be submitted (using a zip archive or a single pdf per main figure if multiple images need to be supplied for one panel). Additional information on source data and instruction on how to label the files are available at .

10) We replaced Supplementary Information with Expanded View (EV) Figures and Tables that are collapsible/expandable online (see examples in <https://www.embopress.org/doi/10.15252/embj.201695874>). A maximum of 5 EV Figures can be typeset. EV Figures should be cited as 'Figure EV1, Figure EV2" etc. in the text and their respective legends should be included in the main text after the legends of regular figures.

12) Our journal encourages inclusion of *data citations in the reference list* to directly cite datasets that were re-used and obtained from public databases. Data citations in the article text are distinct from normal bibliographical citations and should directly link to the database records from which the data can be accessed. In the main text, data citations are formatted as follows: "Data ref: Smith et al, 2001" or "Data ref: NCBI Sequence Read Archive PRJNA342805, 2017". In the Reference list, data citations must be labeled with "[DATASET]". A data reference must provide the database name, accession number/identifiers and a resolvable link to the landing page from which the data can be accessed at the end of the reference. Further instructions are available at .

Further instructions for preparing your revised manuscript:

- a point-by-point response to the referees' comments, with a detailed description of the changes made (as a word file).
- a word file of the manuscript text.

- individual production quality figure files (one file per figure)

- a complete author checklist, which you can download from our author guidelines

(<https://www.embopress.org/page/journal/14602075/authorguide>).

- Expanded View files (replacing Supplementary Information)

We realize that it is difficult to revise to a specific deadline. In the interest of protecting the conceptual advance provided by the work, we recommend a revision within 3 months (29th Dec 2023). Please discuss the revision progress ahead of this time with the editor if you require more time to complete the revisions. Use the link below to submit your revision:

We are sincerely grateful for the insightful comments and critiques provided by Referee #1 and Referee #2. Their invaluable feedback has significantly enhanced the quality and clarity of our manuscript. In this revised version, we have addressed each point raised by the Referees (Referee points are *italicized*). We have included additional experimental data and detailed explanations and made necessary modifications throughout the manuscript (highlighted in blue). We have also added new figures and updated figure numbers accordingly.

Referee #1:

Senoo et al present an in-depth investigation of the association between the ADP/ATP carrier (AAC) and cardiolipin (CL) in yeast. It has been known for some time that AAC binds 3 CL molecules at specific sites (binding pockets). These sites were previously identified by crystallography. The AAC-CL interaction is deemed functionally important because it stabilizes the tertiary structure of the AAC, an abundant mitochondrial protein that is crucial for oxidative phosphorylation.

In order to establish the mechanism of AAC-CL interactions, the authors created several AAC mutants by substituting amino acids in the 3 binding pockets. They FLAG-tagged the mutated AACs, purified them, and analyzed CL-AAC interactions by native mass spectrometry. This is the most interesting and innovative part of the study. As expected, mutations weakened CL-AAC binding but the effects were not specific for the targeted pockets because most mutations (except I151E & L194E) led to the loss of all 3 CL molecules (Fig 2D). The authors went on to study the physiological effects of the mutations, which turned out to be variable. Some mutations slowed down growth. Most mutations reduced ADP/ATP exchange, oxygen consumption, and respiratory control ratios (RCRs). The mutations also altered AAC migration in BN-PAGE from a distinct band to a smear and reduced the amount of AAC that co-migrated with respiratory supercomplexes. Finally, the authors modeled a myopathy-causing mutation of the human adenine nucleotide carrier in yeast and in 293 cells. They found similar effects of this mutation as described for the "binding pocket mutants". Molecular dynamics simulations suggested that the human mutation altered the structure of binding pocket 2, which reduced the likelihood of CL to reach the binding pocket (on-rate) but did not cause the loss of CL that was already in the pocket (off-rate).

The paper describes an impressive body of work but the key conclusions are not supported by the data. In my mind, it will be difficult to develop a clear-cut mechanistic interpretation because of the intricate relation between CL binding and protein conformation. This leaves us with a mostly descriptive set of data.

We thank the reviewer for acknowledging the volume of work performed related to the original submission. In fact, this was a project started in 2010 when my (Steven Claypool) first graduate student (Matt Baile) took a structural modeling class in which he designed a mutagenic strategy to "individually" disrupt the three Aac2-CL interactions revealed in the crystal structures available at that time. Also, we thank the reviewer for challenging us to

overcome the limitations tied to the original submission. We strongly feel that the additional native mass spectrometry and MD analyses performed, and the new characterization of 3 additional transport-impaired Aac2 mutants unrelated to CL binding, transform this study from descriptive to mechanistic and provide the missing elements needed to support its key conclusions. We hope that Referee 1 agrees.

MAJOR CRITIQUE

1. The main problem of the manuscript is that it is difficult, if not impossible, to separate local effects on CL binding from distant effects on secondary and tertiary structure. Each mutation probably altered the local environment, which may have caused the loss of the local CL resident, but in addition, each mutation clearly had long-range effects, which destabilized the binding of distant CL molecules and possibly altered other aspects of the protein structure. Therefore, one cannot associate specific functional defects with the loss of distinct CL molecules. Unfortunately, this undermines the main conclusions of the paper, such as "...that the three buried CLs are functionally distinct..." (results) or that "... transport activity was impaired in a binding site-specific manner" (abstract). For any of the mutations, it is not clear whether the loss of CL is the actual cause of protein dysfunction or merely an associated phenomenon. This raises the question of whether or not mutations outside the binding pockets also affect CL binding. Furthermore, different mutations in the same pocket did not always show the same effect, which further discredits the idea of "pocket-specific functions".

We thank the reviewer for their thoughtful evaluation of our work. The central issue raised here appears to be one of specificity: do the mutations specifically disrupt CL-binding in the targeted pocket or instead have off-target structural effects? In an effort to tackle this important but challenging issue, we implemented a multi-pronged strategy as outlined below:

- a. Do the CL-binding mutations secondarily impact Aac2 protein structure? Based on the following analyses, we consider the impact of the CL-disrupting mutations on protein folding to be minimal.
 - In the original submission, we presented biochemical data, including the mutant's topology in the IMM (Fig. 1E in the revised manuscript) and blue-native PAGE run with CATR and BKA pre-treatments (Fig. 3). These data strongly suggest that the mutants are properly folded with the correct IMM topology prior to detergent extraction and further downstream biochemical analyses (native mass spec, BN-PAGE, or co-IP).
 - We have re-analyzed the MS spectra of the mutants to assess their charge states because charge state perturbations can indicate folded or unfolded states of proteins (Hanauer et al, Nano Lett 2007; Clemmer & Jarrold, J Mass Spectrom 1997). We did not observe a charge state alteration for any of the mutants that was sufficient to consider that its secondary structure was significantly changed relative to wild type

Aac2. Minor structural perturbations, potentially reflecting modest changes to their surface area, were noted for select mutants. We have included representative spectra of each mutant in Appendix fig. S4 and S8 and added these analyses and their discussion to the Results section, “**Specific lipid-protein interactions are disrupted in CL-binding Aac2 mutants**”, as shown below:

“Since the charge state series of a protein in a mass spectrum is related to the exposed surface area of the protein, it is an exquisitely sensitive proxy for the overall fold of the protein (Hall & Robinson, 2012). Evidence for this relationship comes from ion mobility experiments which show that in the gas phase, higher charge states correspond to more unfolded proteins while lower charge states represent more compact structures. It is also evident when there is a particularly large shift in charge state when a protein is unfolded. A minor perturbation in folded structure may result in a slight shift in charge state distribution of one protonation site (Hanauer et al, 2007; Clemmer & Jarrold, 1997). With this in mind, we compared the most abundant charge state 8+ in each case (representative spectra are shown in Appendix fig. S4). In comparison to the WT charge state distribution, there was no large shift in the charge state for all of the mutants investigated, indicating that all the mutants are folded. Several mutants, including L155E, N90E, G172E, R191D, had slight perturbation of their charge states, implying small changes to their surface area may have occurred; however, these changes are not sufficient for the protein to be considered unfolded.”

- To independently see if mutations induce significant structural perturbations, we have conducted 1-microsecond equilibrium MD simulations of hANT1 WT and mutants (L141E and L141F). The results are illustrated in Appendix figs 13-14 in the revised manuscript. The global stability analyses using RMSD (fig. S13A) and alpha-helical content (fig. S14A-B) do not indicate unfolding or large-scale conformational changes in the WT or mutant simulations. Examination of local structural features around pocket 1 and pocket 3, which are distal to the L141 site, indicate high stability, with slight increases in structural fluctuations in the mutated systems. We have added this latter new analysis (fig. S14) and its discussion to the Results section, “**Disturbed CL-ANT1 binding underpins an uncharacterized pathogenic mutation**”, as shown below:

“However, the mutations at pocket 2 do not appear to cause large scale destabilization to the overall protein fold as shown by RMSD analysis (Appendix fig. S13A) or by alpha-helical content (Appendix fig. S14A). Furthermore, the local disruption caused by L141 mutations at pocket 2 does not appear to have significant structural effects on pocket 1 or 3, as analyzed by local RMSD calculations (Appendix fig. S14C-F). In unbound simulations the L141E mutant does display some larger structural fluctuations at both pocket 1 and pocket 3, compared to other

systems, but the increased RMSD is not sustained and at the end of the simulations the RMSD value returns to a value consistent with WT and L141F simulations.”

- b. Do mutations outside the binding pockets also affect CL binding? We have characterized three additional yeast Aac2 mutants, which include a pathogenic variant, A137D (Ogunbona et al, Mol Biol Cell, 2018; PMID: PMC6014099), and two that participate in a transport-related salt-bridge network on the cytoplasmic side, K112D and K215D (PMIDs: 26453935, 30611538, 24474793). These mutations are located outside of CL-binding sites but do impact transport activity. We found that these mutants are functionally compromised, as expected. Critically, each mutant retained CL bindings (native mass spec) and assembly/folding (blue native-PAGE) comparable to WT. Combined, these new results, which are outlined in Fig. 8 and discussed in the newly added “**The ability of CL to stabilize Aac2 tertiary structure is specific**” Results section (attached below), demonstrate that mutations that impact Aac2 transport activity do not necessarily compromise CL binding.

“The ability of CL to stabilize Aac2 tertiary structure is specific.”

Finally, to clarify the causal relationship between disturbed CL binding and AAC dysfunction, we investigated additional yeast Aac2 mutations with documented transport defects (Ogunbona et al, 2018; Palmieri et al, 2005; Ruprecht et al, 2014, 2019; King et al, 2016). Given their location within Aac2 (Fig. 8A), these additional transport-defective mutants are not expected to impact CL bindings which occur at the matrix side (Fig. 2B). We focused on three residues and replaced them with Asp: a previously characterized transport-null pathogenic mutation A137D located in the transmembrane region (Ogunbona et al, 2018; Palmieri et al, 2005) and two transport hypomorphic mutations that disrupt a salt-bridge network on the cytoplasmic side, K112D and K215D (Ruprecht et al, 2014, 2019; King et al, 2016) (Fig. 8A). As expected, the A137D mutant failed to support respiratory growth (Fig. 8B) and was largely devoid of transport activity (Fig. 8C). In contrast, the two salt-bridge mutants retained some, albeit less than WT, respiratory growth (Fig. 8B) and had approximately 50% reduced transport capacity compared to WT (Fig. 8C). Native MS analysis showed that A137D, K112D, and K215D mutants quantitatively associated with 2- or 3- CLs the same as WT (Fig. 8D). This result indicates that these mutations had no impact on CL bindings, in turn demonstrating that CL binding disruption is not attributed to nonspecific or secondary effects of AAC transport dysfunction. Blue-native analysis illustrated that A137D, K112D, and K215D mutants migrated like WT, demonstrating that their tertiary structure is intact (Fig. 8E). Based on these results, we conclude that the ability to stabilize AAC/ANT tertiary structure is a specific function of three evolutionarily conserved CL binding events.”

- c. Why do different mutations in the same pocket not always show the same effect? For the functional analyses, we performed two-way ANOVA analyses, and the results statistically demonstrated pocket-specific differences in ADP/ATP exchange and ADP-stimulated respiration and RCR (Fig. 4E). Yet, these results also depicted that several mutations (N90E for pocket 2, R191D and L194E for pocket 3) presented a slightly different behavior than the others. Among them, N90E and R191D are close to several residues involved in the AAC transport cycle, which was only reported while we conducted this work (PMID: 35739110). We interpret our results as N90E and R191D may exert additional non-CL binding related effects, which we described in the original submission. We have revised the result section “**Engineered CL-binding Aac2 mutant**” which now includes a more thorough discussion of our design rationale, identifies residues that target similar regions of each CL-binding pocket, and clearly distinguishes those that could influence transport-related substrate binding (attached below). Newly added Fig. 1B highlights the placement of each mutation in Aac2.

“The mutations were designed at symmetrically related locations per pocket. One type of mutation is on even-numbered transmembrane helices (I51 for pocket 1, L155 for pocket 2, M255 for pocket 3) and another is on the matrix loops immediately preceding matrix helices (G69 for pocket 1, G172 for pocket 2, G267 for pocket 3) (Fig. 1B). Additional mutations were designed for pocket 2 (N90) and pocket 3 (R191, L194) in a contiguous region of the matrix helices and even-numbered transmembrane helices (Fig. 1B). Predicted to prevent the engagement of the associated CL phosphates like the other mutants, it is of note that N90 and R191 are close to several residues involved in substrate binding related to the AAC transport cycle as recently shown (Mavridou et al, 2022).”

2. Interpretation of the BN-PAGE data is problematic. Conventionally, one thinks of the migration behavior of a protein in a BN-PAGE experiment to either reflect its size and charge or that of complexes it binds to. Based on their BN-PAGE data, the authors concluded that AAC mutants are properly folded in the IMM (correctly I believe) but that the mutations changed the "tertiary fold" of the AAC under BN-PAGE conditions. What does that mean? How can a protein be properly folded but at the same time have the wrong tertiary fold? Similarly confusing is the term "monomeric folding structure". Is this supposed to say the protein is a monomer? My interpretation of the BN-PAGE data is that the mutations make the AAC prone to aberrant interactions with other proteins. This may or may not have anything to do with CL.

The reviewer is absolutely correct in terms of how BN-PAGE data has been traditionally interpreted. However, using Aac2 as a model, we recently established that BN-PAGE can, on a case-by-case basis, additionally report on a protein's tertiary stability and conformation (Senoo et al, Sci Adv 2020, PMID: 32923632). As this was only recently reported, in the revised

manuscript we have clarified our interpretation of the BN-PAGE results and included a discussion of the points outlined below.

- The BN-PAGE migration of Aac2 has been previously discussed extensively in two prior studies from our group (Senoo et al, Sci Adv 2020, PMID: 32923632; Claypool et al, J Cell Biol 2008, PMID: 18779372). Our observations collectively indicate that the migration of AAC at low molecular weight (<67 – 140 kDa) detected following blue native-PAGE reports on the stability of the AAC tertiary structure: in the presence of CL, wild type Aac2 migrates as a properly folded monomer of ~140 kDa; in the absence of CL, the tertiary structure of Aac2 is weakened such that post-solubilization and BN-PAGE resolution, it instead migrates in an unfolded/partially unfolded state. This unfolded/partially unfolded smear can be “rescued/prevented” by pre-incubating CL-lacking mitochondria with AAC inhibitors that only engage properly folded AAC. Our interpretation is that the smear in the low molecular weight range (~60-100 kDa) includes the monomeric Aac2, whose tertiary stability is compromised, likely with lipids and/or Coomassie blue randomly engaged.
- For CL-lacking mitochondria, we previously demonstrated via co-affinity purification that the smear section of Aac2 in the low molecular weight range on blue native-PAGE does not correlate with the accumulation of aberrant interactions between Aac2 and other proteins (Claypool et al, J Cell Biol 2008, PMID: 18779372). Instead of an accumulation of aberrant associations minus CL, the interactions between Aac2 and other mitochondrial proteins detected in the context of CL-containing mitochondria are drastically destabilized in the absence of this mitochondrial phospholipid.
- To expand this prior finding to the panel of CL-binding mutants studied here, we conducted FLAG immunoprecipitation of the CL-binding mutants to experimentally determine their interaction status. Sypro Ruby staining following SDS-PAGE separation of the purified eluates show that the mutants retained identical protein interaction patterns to WT (EV fig. 1), indicating that CL-binding mutants do not robustly engage in aberrant protein interactions. The following paragraph was added to the “**CLs directly bound to Aac2 stabilize its tertiary structure**” Results section:

“To rule out the possibility of aberrant interactions contributing to the smear seen in the CL-binding mutants, we performed FLAG immunoprecipitation and analyzed the pattern of protein interactions by SDS-PAGE (EV fig. 1). The results showed that WT Aac2 was co-detected with multiple bands, with those ranging from 10 to 40 kDa estimated to be the interaction with respiratory complex subunits, as per our previous study (Claypool et al, 2008). The mutants retained identical protein interaction patterns to WT, although N90E displayed relatively weaker interactions, particularly evident in the 20-30 kDa range. These results strongly suggest that mutants do not engage in aberrant protein interactions. This further underscores our interpretation that the low molecular weight

smear detected following blue native-PAGE reflects the compromised tertiary stability of each CL-binding mutant.”

- Additionally, we conducted blue native-PAGE analyses for the three additional Aac2 mutants described above (A137D, K112D, K215D). The migration pattern of the three mutants indicates that their tertiary structure is stable just like WT Aac2 (Fig. 8). Given that their CL-bindings were indistinguishable from WT Aac2, as mentioned in point 1 above, this result emphasizes that the ability to stabilize Aac2 tertiary structure is specific to the binding of CL.

MINOR ISSUES

3. *The abstract does not project a clear message and does not capture the content of the work very well.*

We have revised the abstract with the goal of enhancing its clarity and to more accurately reflect the work. We are always open to suggestions on how to improve in this regard.

4. *Fig 2D is very important. Therefore, individual data should be presented in addition to means and standard deviations.*

We agree that this data is very important and appreciate that the reviewer acknowledges this fact! We have now included individual data points in the bar graph shown in Fig. 2D, as suggested.

5. *The sentence "The respiratory growth capacity of *crd1*Δ and some of the Aac2 mutants was exacerbated at elevated temperature" is probably trying to say that the growth defect was exacerbated at elevated temperature.*

That is absolutely correct. We have revised as suggested.

6. *In the title, the expression "clinically relevant" is odd because the work is scientifically relevant but has no obvious practical implications for the management of patients with AAC mutations. Perhaps "disease-causing" would be better.*

We have revised the title but due to character restrictions, had to remove clinically relevant altogether. Thank you for the "disease-causing" suggestion. We would have used it if space allowed!

Referee #2:

This is an interesting and well-written manuscript. The flow of the experiments is logical, and the experiments are solid. It will make an important contribution to the field.

We thank the reviewer for their supportive view of our work and its potential importance to the field.

1) *How do different CL species affect the function of the AAC? What is the effect of ALCAT and Delta6 desaturase, which were shown to be changed in metabolic disease, affecting respiratory function? I do not ask for additional experiments but to make this study relevant to disease, it will be valuable to consider the different perturbations to CLs.*

This is a great question and one we have thought about extensively. A classic study from the Klingenberg Lab focused on the association between cardiolipin and cow AAC1 concluded that the acyl chain composition of CL, including its saturation status, does not impact its ability to strongly engage the carrier (PMID: 1649052). In support of this conclusion, we showed that yeast Aac2 assembly and function is the same in mitochondria with unremodeled CL (lacking the CL deacylase, Cld1) as with fully remodeled CL (wild type; Baile et al, J Biol Chem, 2014; PMID: 24285538). Interestingly, we did note that in the absence of the monolyso-CL transacylase, TFAZZIN, the amount of Aac2 associated with respiratory supercomplexes is decreased (Baile et al, J Biol Chem, 2014; PMID: 24285538). This observation suggests that Aac2 quaternary assembly either requires a certain threshold of CL or is destabilized by the accumulation of monolyso-CL that occurs when TFAZZIN function is missing. To our knowledge, the effect of ALCAT1, shown by Roger Shi's lab to drive the accumulation of polyunsaturated fatty acid (PUFA)-containing CL species associated with impaired metabolism (PMID: 20674860), and Delta6 desaturase, the rate limiting enzyme in generating long-chain PUFAs which can modulate CL acylation (PMID: 24418793), on mammalian ANT structure and function has not been tested. As both enzymes result in the generation of CL that is more susceptible to oxidative damage, this line of inquiry will be an important goal of our ongoing work in this arena. We appreciate the reviewer's stance that additional experiments are not required to address this point and have attempted to broaden the disease-relevance of our work in the context of the following new paragraph added to the Discussion:

"Moving forward, there are several important avenues to pursue based on our present findings. While the current study focuses on the roles of CL in AAC/ANT's ADP/ATP exchange function, it is worth noting mammalian ANTs have recently been shown to also mediate proton translocation (Bertholet *et al*, 2019). Given that there appears to be a reciprocal relationship between ADP/ATP exchange and proton leak, it will be important in the future to determine the proton transport capacity of human CL-binding ANT1 mutants. Another avenue of future investigation should focus on if and how different CL species differ in their ability to support AAC/ANT structure and activity. A classic study from the Klingenberg

Lab focused on the association between cardiolipin and bovine ANT1 concluded that the acyl chain composition of CL, including its saturation status, does not impact its ability to strongly engage the carrier (Schlame *et al*, 1991). In support of this conclusion, we previously showed that yeast Aac2 assembly and function is the same in mitochondria with unremodeled CL (lacking the CL deacylase, Cld1) as with fully remodeled CL (wild type; (Baile *et al*, 2014)). Interestingly, we did note that in the absence of the monolyso-CL transacylase, TFAZZIN (Schlame & Xu, 2020), the amount of Aac2 associated with respiratory supercomplexes is decreased (Baile *et al*, 2014). As TFAZZIN is mutated in Barth syndrome, a X-linked cardio- and skeletal myopathy, this implicates potential defects in ANT assembly and function in this disease (Barth *et al*, 2004; Clarke *et al*, 2013). Still, the acyl chain complexity of CL in mammals relative to laboratory yeast is vast and includes tissue-specific differences in the final predominant molecular forms of CL and the incorporation of polyunsaturated fatty acids (PUFA) of various lengths (Oemer *et al*, 2020). Incorporation of PUFAs into CL are influenced by the activities of delta 6 desaturase, the rate-limiting enzyme in generating long-chain PUFAs (Mulligan *et al*, 2014), and ALCAT1, an endoplasmic reticulum-resident monolyso-CL acyltransferase (Li *et al*, 2010). PUFA-containing CL species are intrinsically more susceptible to oxidative damage and tied to impaired metabolism (Li *et al*, 2010) and aging (Mulligan *et al*, 2014). To our knowledge, the ability of long-chain PUFA-containing CLs to fully support the structure and function of mammalian ANT, and whether these roles are altered in a meaningful way upon CL peroxidation (Kagan *et al*, 2023), have not been tested and such insight could provide a more detailed understanding of how perturbed CL metabolism contributes to disease pathogenesis.”

2) *Experimentally, the main shortcoming of the approach is the lack of consideration of ATP synthase. How do the authors differentiate effects on CV (ATP synthase) vs. effects on AAC*

The reviewer is absolutely correct that it is difficult to differentiate between unexpected effects on complex V activity versus anticipated effects on AAC structure and function stemming from the mutations inserted into AAC. In the context of an intact mitochondrial inner membrane, their activities are intimately intertwined! In an effort to begin to disentangle predicted consequences of the mutations in terms of AAC function from potential off-target effects on the ATP synthase, we performed in-gel-activity assays for complex V following detergent solubilization and BN-PAGE separation. These analyses strongly suggest that complex V function, at least in reverse (the assay measures hydrolysis and not synthesis) is not changed in the context of any of the CL-binding Aac2 mutants (new EV fig. 4C and Appendix fig S9C). While this new dataset supports the conclusion that the defects observed using functional assays and intact mitochondria reflect changes in mutant Aac2 transport activity, they don't allow us to place the nail in the coffin. As such, in addition to the complex V in-gel-activity assays, we have added a frank discussion outlining the difficulty in cleanly separating the activities of AAC from ATP synthase to the “limitations of this study” section (attached below). In future work that is beyond the scope of the current study, we will be attempting to

reconstitute Aac2 (wild type and mutant forms) in liposomes to isolate AAC function from the influence of ATP synthase or the electron transport chain.

“Limitations of this study

In mitochondria, AAC/ANT-mediated exchange of ADP and ATP, respiratory complex function, and ATP synthesis are intimately integrated in the IMM. Due to this necessary functional coupling, it is challenging to isolate AAC/ANT function from unexpected impacts on respiratory complexes and ATP synthase using intact and thus active mitochondria. As described earlier, AAC transport function is essential for complex IV expression and activity (Ogunbona et al, 2018), and AAC transport-related conformational changes affect protein-protein interactions between Aac2 and respiratory complexes (Senoo et al, 2020). Indeed, we observed measurable respiratory complex defects in Aac2 CL-binding mutants that correlated with the severity of their transport dysfunction, based on the results of complex IV subunit expression (EV fig. 3) and activity (EV fig. 4), Aac2-RSC interactions (Appendix figs. S6, 7), and CCCP-induced respiration rates (i.e. uncoupled respiration from ADP phosphorylation) (EV fig. 2). In accounting for ATP synthesis, a technical limitation is that commonly used methods using mitochondria rely heavily on an exchange of ADP and ATP; for example, the reaction is initiated by adding exogenous ADP to mitochondria and tracked by measuring the produced ATP by luminescence detection (Lanza & Nair, 2009). To isolate complex V function independent of the IMM, we performed complex V in-gel activity assays (EV fig. 4C). While our data indicate that complex V activity of Aac2 CL-binding mutants was intact and therefore unlikely to contribute to the decreases observed in AAC transport activity, this assay detects ATP hydrolysis and not synthesis (Zerbetto et al, 1997). Ultimately, *in vitro* reconstitution into liposomes in the presence or absence of pharmacologically generated membrane potentials are needed to clarify the transport activity intrinsic to each AAC/ANT CL-binding mutant in isolation, an important future goal.”

3) *Is the effect of CL on monomerization specific to CL?*

As commented in response to referee #1's point 1, we have characterized three additional mutants (A137D, K112D, K215D) in which Aac2 transport activity is compromised while CL binding is not. The blue native-PAGE results for these mutants clearly show that their tertiary structure is stable (Fig. 8), which emphasizes that the ability to stabilize Aac2 tertiary structure is specific to the binding of CL.

4) *In Figure 3, I still do not understand how the authors can conclude that the effect of the mutations on the tertiary structure is contributed exclusively and entirely by the CL interaction capacity.*

Related to the reviewer's prior concern (point 3), we investigated three additional Aac2 mutants and provided evidence that Aac2 tertiary structure is not generally impacted by the

insertion of negatively charged amino acids into its sequence. These newly analyzed mutations completely (A137D) or partially (K112D and K215D) disrupt ADP/ATP exchange and yet reside in completely distinct regions of Aac2 relative to its CL-binding pockets. In contrast to the behavior of the three newly analyzed mutants as assessed by BN-PAGE, every single engineered CL-binding Aac2 mutant was structurally destabilized under BN-PAGE conditions unless mitochondria were pre-incubated with AAC inhibitors prior to detergent solubilization. In the net, we feel that our results strongly support the conclusion that disrupting any of the three tightly bound CL molecules weakens, but does not destroy, the tertiary structure of Aac2.

5) Figure 4. Did the authors consider the possibility that the mutants that had reduced exchange velocities changed conformation to become an uncoupling channel, as described by Bertholet et al.? Bertholet et al. indeed find a reciprocal relationship between exchange and uncoupling

This is a very interesting idea and one we have considered. The potential impacts of our mutations on AAC's uncoupling channel function is indeed an exciting possibility. However, it must be noted that Bertholet et al have only shown uncoupling activity for mammalian ANTs. Due to the scope of our current work, which focused primarily on AAC's ADP/ATP transport function, we did not design our experiments to address its uncoupling channel activity. Thus, we have not tested the possibility that any of the mutations analyzed altered the reciprocal nucleotide exchange/proton coupling relationship. As this is an interesting and important point, the CL-AAC interactions and their possible significance for the uncoupling function of AAC has been added to the Discussion section (see Referee 2's point 1).

6) In Figure S4, what is the explanation for the high level of variability in this mutant?

Unfortunately, we do not have a definitive explanation for this variability; however, we can offer some observations. The unenergized (-Mal/Pyr) condition required a higher [ADP] range for the reaction, suggesting slower kinetics compared to the energized condition. We speculate that measurements are more variable in this dynamic range.

7) In Figure 4, the authors test the effect of energized as compared to non-energized mitochondria, which is expected to affect membrane potential. However, the uncoupling ability of the AAC is also expected to change membrane potential. Should the study compare membrane potential and uncoupling in the different mutants?

We recognize that the close relationship between AAC/ANT, complex V, and the respiratory chain makes it difficult to separate their contributions definitively. We have included a discussion about these limitations in the revised manuscript. For the points regarding AAC/ANT versus respiratory complexes, we have clarified several points as described below:

- We have investigated complexes III and IV activities in our CL-binding mutants. The results show that in particular, complex IV activity is impaired in several mutants, which nicely correlates to the extent of their transport defects (Fig. 4 and Appendix figs S5-6). This is in line with our prior determination that AAC transport supports complex IV expression via modulating translation of its mitochondrial DNA-encoded subunits (PMCID: PMC6014099).
- Given the results of complexes III and IV activities, we have extended our interpretation of the CCCP-induced respiration and complex IV expression results presented in the original manuscript. For yeast Aac2, the results suggest measurable respiratory chain damage for pocket 2 mutants in particular. Based on our previous study showing that Aac2 transport dysfunction compromises complex IV expression (PMCID: PMC6014099), we interpret these findings as evidence that Aac2 transport dysfunction can contribute to respiratory chain damage.

These new complex III, IV and V results are now discussed in the “***Disturbed CL-Aac2 interaction attenuates respiratory complex expression, activity, and assembly***” Results section as follows:

“We found that the activity of complex III was mostly unaffected in Aac2 CL-binding mutants, with the exception of pocket 2 mutants, which showed slightly decreased activity (EV fig. 4A). In contrast, complex IV activity was impaired and correlated with the mutants' transport ability (EV fig. 4B). Consistent with our previous finding (Ogunbona et al, 2018), these results indicate that AAC transport dysfunction diminishes the expression and activity of respiratory complexes, particularly complex IV; this likely accounts for the reduced CCCP-induced respiration for CL-binding mutants with AAC transport dysfunction (EV fig. 2A-B). The results reinforce our conclusion that interactions between CL and Aac2 are crucial for AAC transport function. The mutants still maintained the expression of nuclear genome-encoded subunits of respiratory complexes (complex III, complex IV, and complex V) and the mitochondrial DNA-encoded subunit of complex V, Atp6 (EV fig. 3), as well as complex V activity (EV fig. 4C).”

- For human ANT1 mutants, we showed in the original submission that maximal respiration of the L141E mutant induced by FCCP (Fig. 6F) was reduced compared to WT or the L141F mutant. In search of a mechanism, we investigated the steady-state expression of respiratory complex subunits; however, we only observed a decrease in a complex I subunit in the L141F mutant. Thus, in contrast to yeast mutants, a correlation between ANT function and respiratory complex damage was unclear in the human cell model. Clarifying potential species-specific effects would require further investigation.

As mentioned in point 5 above, the experimental systems employed in the current study do not allow us to draw any conclusions related to the uncoupling function of AAC. If and how ANT1-CL binding disruption impacts proton translocation is an important line of investigation, but one that is beyond the scope of the current study.

8) *The statement that ADP-stimulated respiration is a reflection of AAC activity is incorrect. A linear relationship between respiration and ADP availability is limited to a situation where respiration is not limited by the respiratory chain complexes or substrate.*

Thanks for this correction. As mentioned in response to point 7, our results in yeast indicate that the transport dysfunction in our CL-binding mutants is linked to changes in the expression and function of respiratory complexes, in particular at the level of complex IV. This limitation of the current study is included in the revised manuscript. Worth considering is that a correlation between reduced mutant ANT transport and impaired respiratory complex expression and function was not observed suggesting that the same might be true for the yeast mutants if Aac2 transport and complex IV expression could be cleanly separated. Future *in vitro* reconstitution studies are being pursued to overcome this current limitation.

9) *Fig S5 indicates that similar trends of mutant effects were found with CCCP. However, in the presence of CCCP, the AAC does not play any role in determining respiratory rate. Isn't that an indication that the mutants affect the respiratory chain independent of ADP/ATP transport?*

Similar to the above points 7 and 8, our results indicate that AAC transport dysfunction in our CL-binding mutants is linked to changes in the expression and function of respiratory complexes, in particular at the level of complex IV. Based on these results, we have contextualized our interpretation of the CCCP-induced respiration presented in the original manuscript.

10) *Authors should at least mention previous studies demonstrating the activity of AAC as a proton transport channel, as shown by Berthollet et al. If authors consider that H⁺ is also transported by AAC, it will be very interesting to measure or discuss the implications of this activity in their system with their mutants. This could also affect the interpretation of the respiratory experiment, as shown in Figure 6F.*

Similar to the response to point 5, although we acknowledge the multifaceted function of AAC/ANT, our current study did not address the proton transport activity or uncoupling function of AAC due to its scope. The possible contribution of CL-AAC interactions on AAC proton transport has been added to the Discussion section. And future work is being planned to directly measure proton flux mediated by the CL-binding mutants, potentially in collaboration with the Bertholet Lab.

Dear Steve,

Thank you for submitting the revised version of your manuscript, which addresses the concerns of the referees. This revised version has now been re-reviewed; I attach the second referee reports to the bottom of this mail. As you will see, you have addressed the referees' concerns to their satisfaction.

Before I can formally accept your manuscript for publication, however, there are some remaining editorial points which need to be addressed. In this regard would you please:

- rename the Conflict of Interest statement the 'Disclosure and Competing Interests Statement',
 - remove the author credit section from the manuscript,
 - rename the callouts for EV figures as Figure EV1-EV5, instead of EV fig. 1-5,
- when unzipped, figure files 1, 2 and 4 are more than 300MB and should be resized,
- upload the appendix file in PDF format; the title should be stated on the first (table of contents) page of the appendix,
 - save Source Data in a scheme of one figure per folder and then uploaded as .zip files; i.e. all the Source Data files for figure 1 need to be saved in a single folder and this needs to be zipped and then uploaded as "SD figure 1.zip" file,
 - consider including a separate 'Data Information' section in the legends of figures 2a-b; 4a-b, e-f,
 - define the annotated p values ****/††††/****/†††/**/††/*/† in the legend of figure 5b and 8g as appropriate,
 - define box-plot in terms of minima, maxima, centre, bounds of box and whiskers in the legend of figure 1c,
 - define error bars in the legends of figures 2d, 5b, 7e and 8f-g
 - define n in the legends of figures 2d, 5b, 7e and 8f-g,
 - check the give email address for co-author Teona Munteanu,
 - correct section order as follows: title page with complete author information, abstract, keywords, introduction, results, discussion, materials & methods, data availability section, acknowledgements, disclosure and competing interests statement, references, main figure legends, tables, expanded figure legends,
 - indicate the statistical test used for data analysis in the legends of figures 5b and 8g, and
 - in figures 2d, 5f-g and 8d-e there is a mismatch between the annotated p values in the figure legend and the annotated p values in the figure file that should be corrected.

We include a synopsis of the paper (see <http://emboj.embopress.org/>). Please provide me with a two-sentence summary together with 3-5 bullet points that capture the key findings of the paper.

We also need a summary figure for the synopsis. The size should be 550 wide by [200-400] high (pixels). You can also use something from the figures if that is easier.

EMBO Press is an editorially independent publishing platform for the development of EMBO scientific publications.

Best wishes,

William

William Teale, PhD
Editor
The EMBO Journal
w.teale@embojournal.org

- a point-by-point response to the referees' comments, with a detailed description of the changes made (as a word file).
- a word file of the manuscript text.
- individual production quality figure files (one file per figure)
- a complete author checklist, which you can download from our author guidelines (<https://www.embopress.org/page/journal/14602075/authorguide>).

- Expanded View files (replacing Supplementary Information)

We realize that it is difficult to revise to a specific deadline. In the interest of protecting the conceptual advance provided by the work, we recommend a revision within 3 months (22nd Jul 2024). Please discuss the revision progress ahead of this time with the editor if you require more time to complete the revisions. Use the link below to submit your revision:

Referee #1:

The authors produced another revision, in which they addressed the criticism of two reviewers. The revision is an honest attempt to tackle the complicated issue of how bound lipids affect protein structure and vice versa. Overall, the manuscript contains a wealth of information and has pushed this topic as far as one can go with the available technologies. While the paper may provoke controversial discussions, I believe those discussions should be held within the scientific community rather than between authors and reviewers. I therefore recommend to publish this work without further changes.

Referee #2:

The manuscript has been significantly improved.
My comments were addressed.
I have no further comments.

- rename the Conflict of Interest statement the 'Disclosure and Competing Interests Statement'.
 - Done.
- remove the author credit section from the manuscript.
 - Removed "Author contributions" section.
- rename the callouts for EV figures as Figure EV1-EV5, instead of EV fig. 1-5. when unzipped, figure files 1, 2 and 4 are more that 300MB and should be resized
 - Corrected the callouts for EV figures.
 - Figures 1, 2 and 4 have been resized.
- upload the appendix file in PDF format, the title should be stated on the first (table of contents) page of the appendix
 - Done.
- save Source Data in a scheme of one figure per folder and then uploaded as .zip files; i.e. all the Source Data files for figure 1 need to be saved in a single folder and this needs to be zipped and then uploaded as "SD figure 1.zip" file
 - Done.
- consider including a separate 'Data Information' section in the legends of figures 2a-b; 4a-b, e-f,
 - We were unsure exactly how to do this. If this is deemed important, some guidance as to what you are envisioning would be welcome.
- define the annotated p values ****/++++/***/+++/**/+/ in the legend of figure 5b and 8g as appropriate
 - Corrected for 5b and 8g
- define box-plot in terms of minima, maxima, centre, bounds of box and whiskers in the legend of figure 1c
 - Done.
- define error bars in the legends of figures 2d, 5b, 7e and 8f-g
 - 2d, 8f: nothing has changed since they were already indicated or not applicable.
 - 5b, 7e, 8g: Corrected.
- define n in the legends of figures 2d, 5b, 7e and 8f-g
 - 2d, 8f: nothing has changed since they were already indicated.
 - 5b, 7e, 8g: Corrected.
- check the give email address for co-author Teona Munteanu
 - tmunteanu27@gmail.com

- correct section order as follows: title page with complete author information, abstract, keywords, introduction, results, discussion, materials & methods, data availability section, acknowledgements, disclosure and competing interests statement, references, main figure legends, tables, expanded figure legends
 - Corrected.
- indicate the statistical test used for data analysis in the legends of figures 5b and 8g
 - Included.
- in figures 2d, 5f-g and 8d-e there is a mismatch between the annotated p values in the figure legend and the annotated p values in the figure file that should be corrected.
 - 2d: nothing has changed since there was no mismatch.
 - 5f-g, 8d-e: Corrected.

Dear Steve,

I am pleased to inform you that your manuscript has been accepted for publication in the EMBO Journal.

I am really happy to see this work in The EMBO Journal. Congratulations to you and your team!

Best wishes,

William

William Teale, PhD
Editor
The EMBO Journal
w.teale@embojournal.org
